# Direct asymmetric synthesis of β-branched aromatic α-amino acids using engineered phenylalanine ammonia lyases

Chenghai Sun [1] ✉, Gen Lu[2], Baoming Chen[2], Guangjun Li[2], Ya Wu[2], Yannik Brack [1], Dong Yi[3], Yu-Fei Ao [1], Shuke Wu [4], Ren Wei [1], Yuhui Sun[2], Guifa Zhai [2] ✉ & Uwe T. Bornscheuer [1] ✉

β-Branched aromatic α-amino acids are valuable building blocks in natural products and pharmaceutically active compounds. However, their chemical or enzymatic synthesis is challenging due to the presence of two stereocenters. We design phenylalanine ammonia lyases (PAL) variants for the direct asymmetric synthesis of β-branched aromatic α-amino acids. Based on extensive computational analyses, we unravel the enigma behind PAL's inability to accept β-methyl cinnamic acid (β-MeCA) as substrate and achieve the synthesis of the corresponding amino acids of β-MeCA and analogs using a double (PcPAL-L256V-I460V) and a triple mutant (PcPAL-F137V-L256V-I460V). The reactions are scaled-up using an optimized *E. coli* based whole-cell biotransformation system to produce ten β-branched phenylalanine analogs with high diastereoselectivity (dr > 20:1) and enantioselectivity (ee > 99.5%) in yields ranging from 41-71%. Moreover, we decipher the mechanism of PcPAL-L256V-I460V for the acceptance of β-MeCA and converting it with excellent stereoselectivity by computational simulations. Thus, this study offers an efficient method for synthesizing β-branched aromatic α-amino acids.

β-branched aromatic α-amino acids are an important class of non-proteinogenic amino acids, which have been found in various natural products and bioactive compounds (Fig. 1a)[1–6]. Although several studies have demonstrated that adding β-substituents can significantly enhance the biological activity of the corresponding parent molecules[7–10], the incorporation of these substituents is still challenging due to the presence of two consecutive chiral centers in β-branched α-amino acids. To date, only two chemical approaches have been developed to synthesize β-branched α-amino acids. One approach involved the asymmetric hydrogenation of β,β-disubstituted didehydroamino acids[11], while the other one was direct asymmetric C-H activation using aliphatic quinolyl carboxamide analogs as starting materials[12,13]. However, the limitations of these chemical asymmetric synthesis constrained their widespread adoption in industry, especially because of the excessive usage of rare transition-metal catalysts, the requirement of pre-installed directing groups, and the suboptimal diastereoselectivity.

In comparison with chemical synthesis, biocatalysis has played an increasingly pivotal role in the synthesis of chiral compounds, owing to its mild reaction conditions and exceptional stereoselectivity[14–17]. In recent years, several enzymatic processes have been developed to produce β-branched α-amino acids, including: (i) engineering of tryptophan synthases (TrpBs) to generate β-alkyl tryptophan analogs[18]; (ii) engineering of methylaspartate ammonia lyases (MALs) for the

¹Department of Biotechnology and Enzyme Catalysis, Institute of Biochemistry, University of Greifswald, Greifswald, Germany. ²School of Pharmacy, Tongji Medical College of Huazhong University of Science and Technology, Hubei Key Laboratory of Natural Medicinal Chemistry and Resource Evaluation, Wuhan, China. ³Research Center for Systems Biosynthesis, China State Institute of Pharmaceutical Industry, National Key Laboratory of Lead Druggability Research, Shanghai, China. ⁴College of Life Science and Technology, Huazhong Agriculture University, Wuhan, China. ✉e-mail: chenghai.sun23@outlook.com; gfzhai@hust.edu.cn; uwe.bornscheuer@uni-greifswald.de

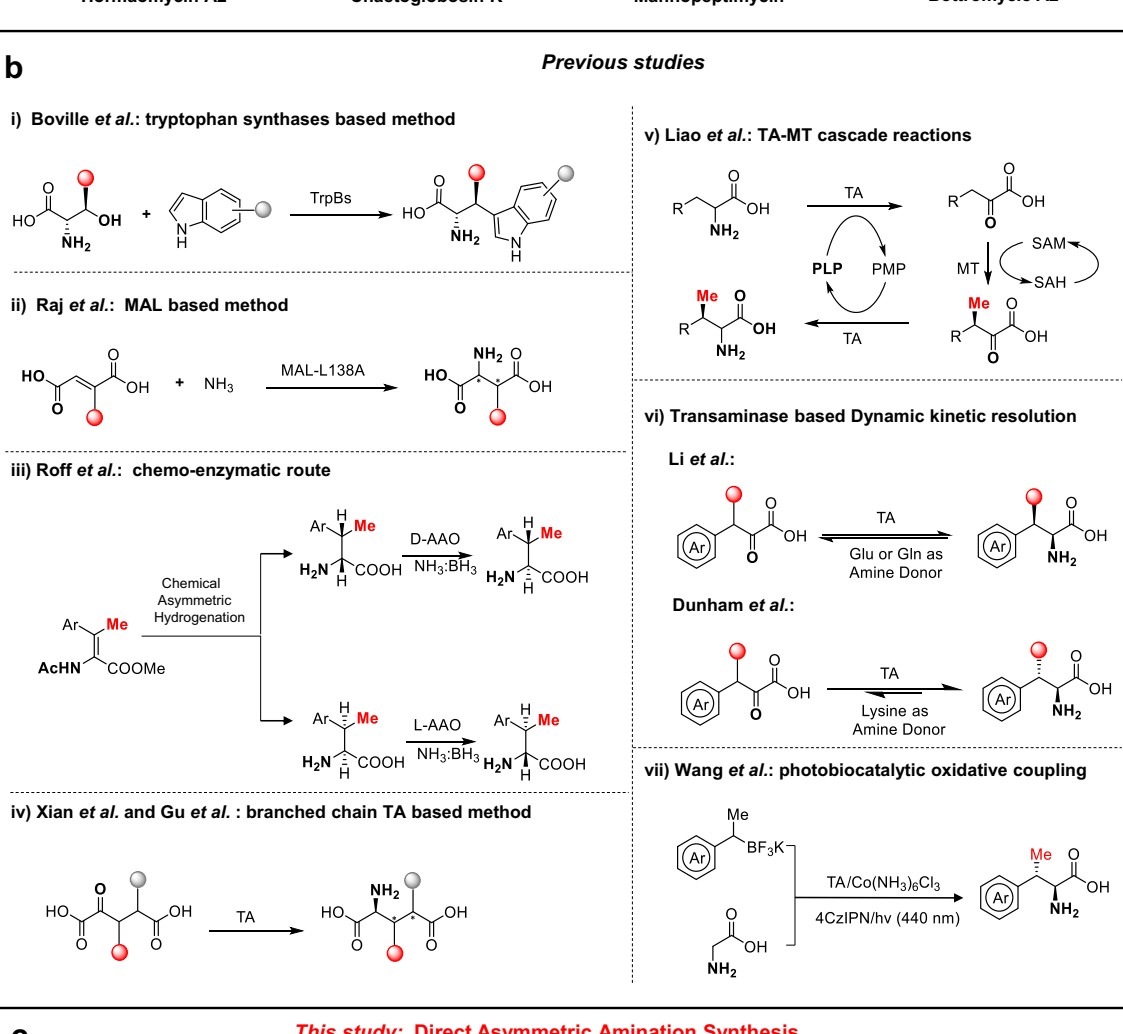

**Fig. 1 | Representative natural products and strategies for synthesizing β-branched aromatic amino acids. a** Examples of bioactive compounds containing β-branched aromatic amino acids. **b** Reported[18–26] strategies for the biocatalytic synthesis of β-branched aromatic amino acids. **c** Direct asymmetric amination synthesis method developed in this study. MAL methylaspartate ammonia lyase,

AAO amino acid oxidase, TA transaminase, PLP pyridoxal-5'-phosphate, PMP pyridoxamine-5'-phosphate, MT methyltransferase, SAM S-Adenosylmethionine, SAH S-adenosylhomocysteine, THF Tetrahydrofuran, TEPA Triethylphosphonoacetate, PcPAL Phenylalanine ammonia lyase from *Petroselinum crispum*.

synthesis of β-branched aspartate derivatives[19]; (iii) chemo-enzymatic approaches employing amino acid oxidases (AAOs) to obtain enantiomerically pure β-methylated α-amino acids[20]; (iv) utilization of transaminases (TAs) for the production of branched glutamate analogs[21,22]; (v) integration of cascade reactions involving TAs and methyltransferases to produce β-methylated α-amino acids[23]; (vi) implementing dynamic kinetic resolution techniques based on TAs[24,25] and (vii) photobiocatalytic oxidative coupling method to prepare β-methylphenylalanines[26] (Fig. 1b). Despite this progress propelled by the above concepts, some of them are hampered by the need of (light-sensitive) pyridoxal-5′-phosphate (PLP) or S-adenosylmethionine (SAM) requiring a complex cofactor regeneration system. Although no co-factors are required for the reactions catalyzed by MALs, these enzymes often exhibit inadequate diastereoselectivity[19].

Compared to the above biosynthetic reactions, a direct asymmetric amination of β-branched cinnamic acid (CA), and its analogs in an anti-Michael regioselectivity way, is much more attractive to synthesize β-branched aromatic α-amino acids. In principle, phenylalanine ammonia lyases (PALs, EC 4.3.1.24) possess the unique capability to catalyze non-oxidative and reversible amination reactions using CA-like substrates[27–30]. Thus, they stand out as promising enzymes for such anti-Michael amination reactions. Besides, PALs can intrinsically generate the catalytic prosthetic group 3,5-dihydro-5-methylidene-4H-imidazol-4-one (MIO) through autocatalysis, eliminating the need for cofactors and cofactor regeneration systems. These features collectively render PALs particularly valuable in industrial applications for the synthesis of phenylalanine analogs, which has been evidenced by several leading companies, including Pfizer (USA)[31], DSM (Netherlands)[32], Novartis (Switzerland)[33], and Johnson Matthey (UK)[34].

In previous reports, PALs from *Rhodotorula glutinis* (RgPAL)[35–37], *Rhodosporidium toruloides* (RtPAL)[38], *Petroselinum crispum* (PcPAL)[39], and *Anabaena variabilis* (AvPAL)[40,41] have been well studied, especially including the engineering of their hydrophobic binding pockets to expand their substrate scope[42–47]. However, in these investigations, only CA analogous with single or multiple substituents exclusively present at the phenyl ring could be converted, including heterocyclic systems, bulkier aryl systems, or naphthalene rings[36,43,48]. An early study in 1994 demonstrated that β-deuterated phenylalanine derivatives could be converted in PAL-mediated deamination reactions[49]. Subsequently, Turner et al. achieved the amination reaction using AvPAL with β-deuterated cinnamic acid[50]. Although this reaction resulted in a low enantiomeric excess (ee) value, it provided evidence that β-deuterated substrates are accepted by PALs. However, for other substitutions apart from deuterium, such as fluorine and methyl substitutions, no conversion was observed with eukaryotic RgPAL[36], bacterial AvPAL[24], or the corresponding amino acid mutase (PAM)[51]. These results underscore the substantial challenges associated with utilizing PALs for the amination of CA analogs bearing β-substituents such as alkyl.

Here, we report a computational analysis based rational design approach to decipher the underlying reason for the inability of PALs to accept β-methyl-cinnamic acid (β-MeCA) as substrate. This leads to the creation of two efficient PAL mutants (PcPAL-L256V-I460V and PcPAL-F137V-L256V-I460V) which catalyze the direct asymmetric amination of various β-MeCA analogs with high diastereoselectivity (dr > 20:1) and enantioselectivity (ee > 99.5%) (Fig. 1c). This work expands not only the substrate scope of PALs, but especially provides a direct reaction pathway for the synthesis of β-branched aromatic α-amino acids.

## Results

### In silico analyses of PcPAL towards β-MeCA

Currently, the E1cb (via a carbanion intermediate) and the E2 (concerted) mechanisms are widely accepted for PcPAL-mediated deamination reactions[27]. Consequently, the amination reactions catalyzed by PcPAL are likely to proceed through the reverse E1cb and E2 mechanisms (Fig. S1). Despite an ongoing debate between these

mechanisms, both of them involve two critical steps in the amination reactions: amination and protonation. The key residues involved in these processes include the MIO-NH$_2$ adduct, Tyr110, and R354. Among them, the MIO-NH$_2$ adduct is responsible for amidating the αC of the substrate, Tyr110 is responsible for protonating the βC, and R354 forms strong hydrogen bonds with the carboxyl group of the substrate to position the substrate. Based on these catalytic processes, we conducted a series of in silico analyses to elucidate the underlying reasons for the inability of PALs to recognize β-substituted substrates. Firstly, CA and β-MeCA were individually docked into PcPAL (PDB:6HQF)[52] (Fig. S2a), and based on the docking results, we further performed conventional molecular dynamics (cMD) simulations. The results showed that in the (β-MeCA)/(PcPAL-WT) complex, the distance variation range between the N atom of MIO and the αC of β-MeCA was narrower than that in the (CA)/(PcPAL-WT) complex (Figs. S3a, b). Besides, by examining the hydrogen bond interactions between the substrates and amino acids within the active pocket (Source Data 3), it was found that in the (β-MeCA)/(PcPAL-WT) complex, the carboxyl group of β-MeCA forms a high-frequency hydrogen bond with Tyr110, appearing in 99% of the recorded 5000 snapshots (Source Data 3). These evidences suggest that β-MeCA is more stable within the pocket. Although the reason for the inactivity of the (β-MeCA)/(PcPAL-WT) complex remains unclear, based on the above observations, we reasonably speculate that the inactivity is likely due to the perturbation of the substrate positioning dynamics[53,54].

To validate our hypothesis, we further carried out umbrella sampling (US) simulations to determine the optimal amination distances (distance 1 in Fig. S4) when CA and β-MeCA have the lowest energy conformations in PcPAL-WT. For substrate CA, the optimal amination distance ranged from 3.5 to 3.9 Å when it had the lowest energy conformations in PcPAL-WT, whereas the corresponding distance ranged from 3.6 to 3.7 Å for β-MeCA (Fig. S2b). Besides these different distance ranges detected in the presence of CA and β-MeCA, we also observed differences related to the angle between MIO and the double bond of the substrates, which is important for the amination process. Specifically, we observed two angles in the (CA)/(PcPAL-WT) complex (70° and 109°, Fig. 2a). However, in the (β-MeCA)/(PcPAL-WT) complex, only one angle was detected around 80° (Fig. 2b). According to the Bürgi-Dunitz angle theory[55], the N atom of MIO should attack from the direction of 105 ± 5° relative to the C=C bond to achieve effective orbital overlap for C-N bond formation. This missing bond-forming angle, along with the relatively narrow optimal amination distance range detected in the (β-MeCA)/(PcPAL-WT) complex, suggests that β-MeCA is less flexible and more stable than CA. Further, we compared the pocket volumes in the presence of both CA and β-MeCA. The results showed that the mean pocket volumes of the (CA)/(PcPAL-WT) complex and the (β-MeCA)/(PcPAL-WT) complex were 192.8 Å$^3$ and 177.4 Å$^3$, respectively (Fig. 2c, d). The smaller pocket volume within the (β-MeCA)/(PcPAL-WT) complex indicates that β-MeCA was confined to a very narrow cavity, which restricts its dynamic changes. Overall, all the US simulation results observed in the (β-MeCA)/(PcPAL-WT) complex indicated that the presence of β-methyl substitution may disrupt the normal substrate positioning dynamic.

Following, Quantum Mechanics/Molecular Mechanics (QM/MM) simulations were employed to further optimize the conformations of PcPAL-WT and the substrate complexes. The QM/MM simulation results showed that, compared to the CA substrate (Fig. 2e) a strong hydrogen bond was detected between the hydroxyl hydrogen of Tyr110 and the carboxyl group of β-MeCA (1.51 Å, Fig. 2f). The formation of this strong hydrogen bond is thermodynamically unfavorable in the enzymatic reaction of PcPAL-WT, not only restricting the flexibility of β-MeCA in the active pocket but also significantly impeding the proton transfer process from Tyr110 to the βC-carbon atom.

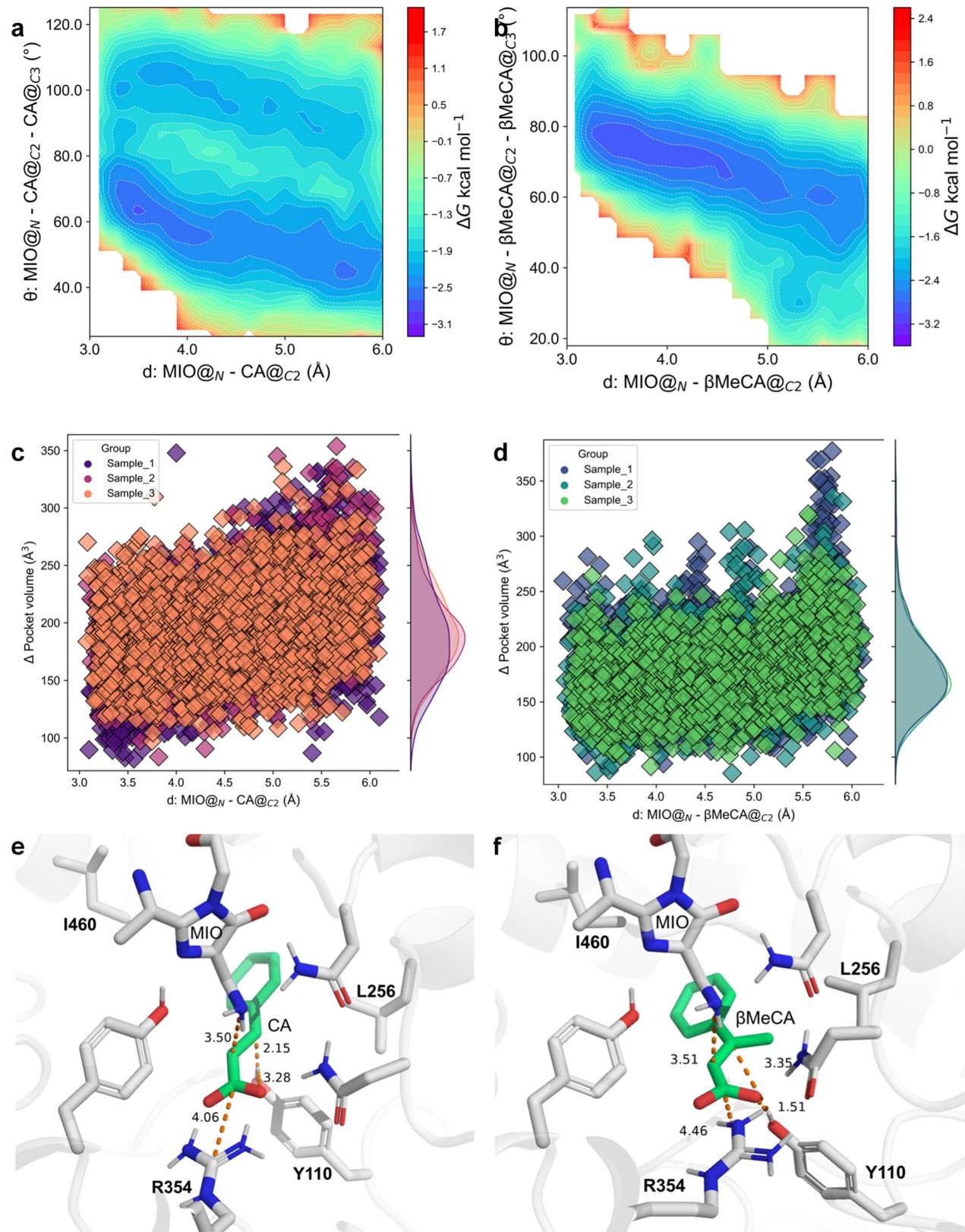

**Fig. 2 | In silico analyses for the substrate recognition mechanism of PcPAL.**
**a**, **b** Gibbs energy diagram of the cMD simulations projected onto the distance from the amino N of MIO to the αC of the substrates (X-axis) and the angle between MIO and the double bond of the substrates (Y-axis) within the (CA)/(PcPAL-WT) complex and the (β-MeCA)/(PcPAL-WT) complex, respectively.
**c**, **d** the pocket volumes detected during the cMD simulations of (CA)/(PcPAL-WT)

complex and the (β-MeCA)/(PcPAL-WT) complex, respectively. **e**, **f** Optimized pre-reaction states of CA and β-MeCA in the PcPAL-WT using the Quantum Mechanics/Molecular Mechanics (QM/MM) methodology, respectively. Source data are provided in Source Data 3 to Source Data 5. CA: cinnamic acid; β-MeCA: β-methyl cinnamic acid; MIO: 3,5-dihydro-5-methylidene-4H-imidazol-4-one.

In summary, our computational simulations indicated that the incapacity of PcPAL-WT to aminate β-MeCA raised from the perturbation of β-MeCA's positioning dynamic within the active pocket, which is primarily due to the strong hydrogen bond formation between the carboxyl group of β-MeCA and Tyr110. Consequently, we hypothesized that alleviating the stringent active pocket of PcPAL might help restore the correct positioning dynamic of β-MeCA analogs to achieve the asymmetric amination for the synthesis of β-branched amino acids. Besides, our computational results indicated that to induce reactive conformations, it is essential to engineer not only the residues around the β-methyl group of β-MeCA, but also those around its phenyl ring. This is crucial as both sets of residues were essential to form intricate interactions with β-MeCA and influence its conformation in the active pocket.

### Rational engineering of PcPAL for the amination of β-MeCA

As the above computational results provided us with understandings for the rational engineering of PcPAL, we identified eight key residues around the substrate binding pocket (L134, F137, L138, I460, E484, L256, N260, and L206), which could affect the correct conformation of β-MeCA (Fig. 3a). Among them, L256 and N260 are located near the β-Me group, while the others are around the phenyl ring of β-MeCA. Subsequently, each of these residues was individually substituted with three smaller residues (A, G, and V). Additionally, we also investigated three other mutants, previously established in our laboratory (E484D, F137V-I460V[43], and F137A-I460A[43]) to explore their potential. Notably, the latter two mutants, developed by the Bencze's group[43], were investigated for their ability to catalyze the amination of bulky substrates, such as styrylacrylate and biarylalanines. After recombinant expression in *E. coli* BL21 (DE3) and purification of these 27 mutants, an in vitro screening against β-MeCA was performed. Remarkably, after reacting with three mutants, i.e., L256V, I460V, and E484D, the product concentrations in the reaction system achieved 24.3%, 8.7%, and 5.3%, respectively. Interestingly, L256V and I460V mutants were well studied previously for synthesizing various L-phenylalanine analogs[43–45,56,57], yet their ability to convert β-substituted CA analogs was not explored yet. Through analyzing the conformation of β-MeCA within the docking result, we found that L256 is positioned in close proximity to the β-methyl group of the substrate (Fig. 3a). After substitution with Val, more space becomes available to accommodate the β-methyl group. This is a

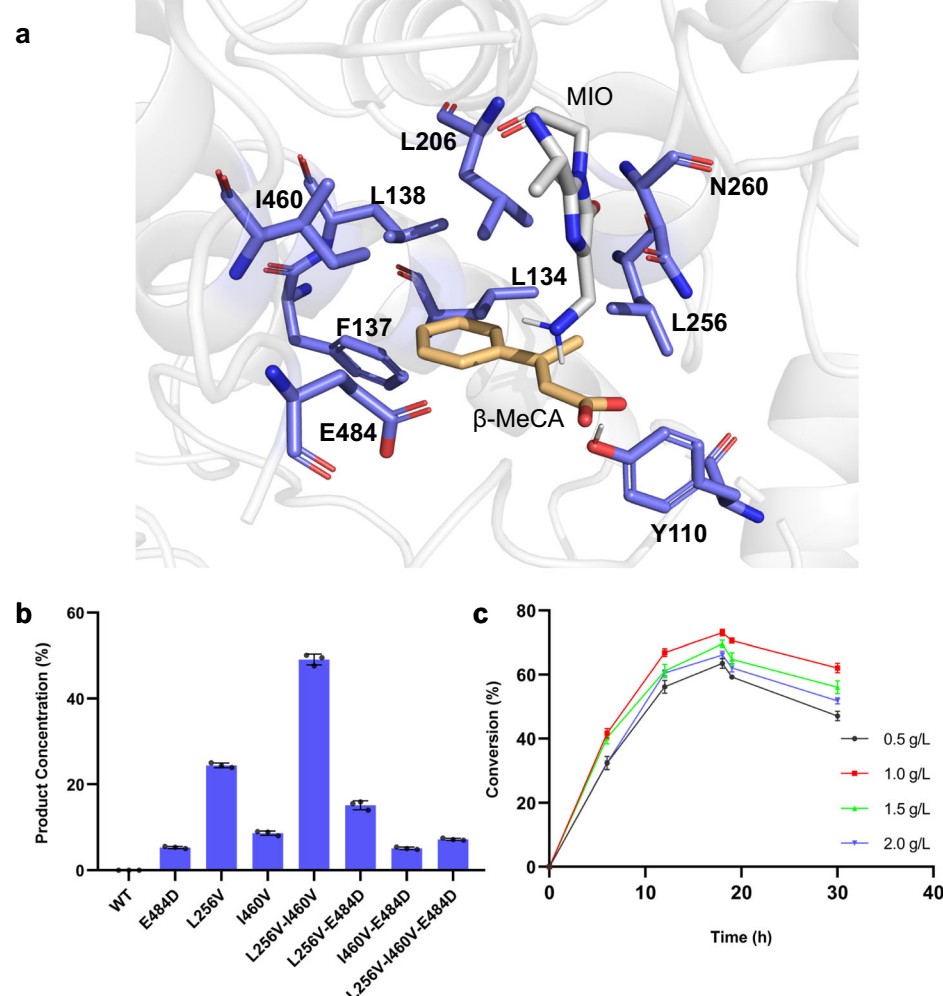

**Fig. 3 | Results of engineering and optimizing whole-cell catalysis conditions for PcPAL in β-MeCA amination. a** Key bulky residues around the substrate β-MeCA modeled in PcPAL-WT. The active site illustrations are based on the crystal structure of PcPAL (Protein Data Bank ID: 6HQF). The substrate and MIO group are highlighted in orange and gray, respectively. **b** Enzyme activity of PcPAL mutants towards β-MeCA. Reaction conditions: 100 μg purified enzyme, 1 mM β-MeCA, total 200 μl in NH₄OH buffer (pH 10.0), 30 °C, 450 rpm for 24 h. After that the reaction was stopped by adding HCl and centrifuged for further HPLC analysis. **c** Time curves with different substrate concentrations. Three parallel samples were collected from each reaction for HPLC analysis. Product concentrations were calculated using a standard curve and the peak area of the product. Error bars (SD) were derived from triplicate experiments and created using GraphPad Prism 8. Data are presented as mean values ± SD. Source data are provided in Source Data 1. β-MeCA: β-methyl cinnamic acid; MIO: 3,5-dihydro-5-methylidene-4H-imidazol-4-one.

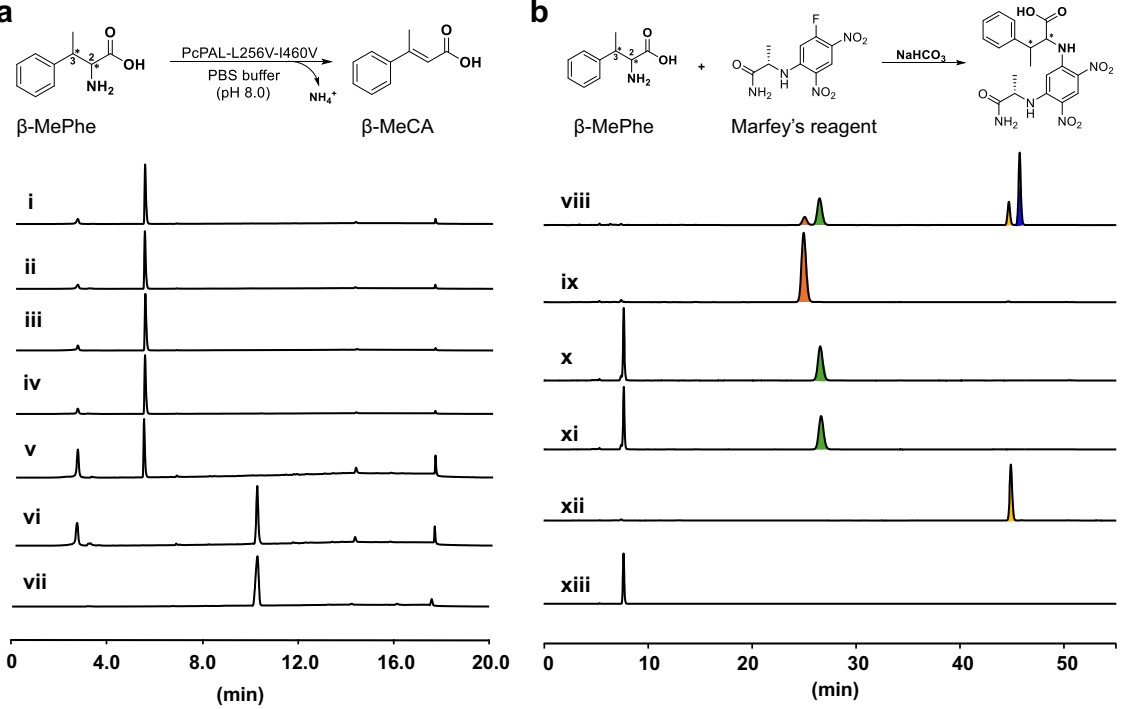

**Fig. 4 | HPLC data for determining the absolute configuration of the amination product. a** In vitro deamination assays with β-MePhe of different absolute configurations using the PcPAL-L256V-I460V mutant. Reaction conditions: 100 μg purified enzyme, 1 mM various β-MePhe, total 500 μL in PBS buffer (pH 8.0), 30 °C, 450 rpm for 24 h. i) (2S, 3S)-β-MePhe; ii) deamination reaction of (2S, 3S)-β-MePhe; iii) (2R, 3R)-β-MePhe; iv) deamination reaction of (2R, 3R)-β-MePhe; v) (2S, 3R)-β-MePhe; vi) deamination reaction of (2S, 3R)-β-MePhe; vii) β-MeCA standard. **b** Derivatization results with β-MePhe of different configurations and the in vitro

reaction product formed by using Marfey's reagent. viii) standard mixture of β-MePhe derivatized with Marfey's reagent; ix) (2S, 3S)-β-MePhe derivatized with Marfey's reagent; x) (2S, 3R)-β-MePhe derivatized with Marfey's reagent; xi) derivation of the in vitro amination reaction product produced by the PcPAL-L256V-I460V mutant; xii) (2R, 3R)-β-MePhe derivatized with Marfey's reagent; xiii) blank containing solely Marfey's reagent and enzyme. β-MePhe β-methyl phenylalanine, β-MeCA β-methyl cinnamic acid.

plausible explanation for the significantly higher activity exhibited by the L256V mutant. I460 and E484 are located around the phenyl group of β-MeCA. After turning them into smaller residues, we thought that the substrate might undergo spatial displacement within the active pocket. This should lead to the generation of additional conformations, among which favorable ones for the reaction may be present, thereby demonstrating reaction activity. To find much better performing variants, we further combined these promising mutants to generate four additional mutants (L256V-I460V, L256V-E484D, I460V-E484D, and L256V-I460V-E484D). Among them, the L256V-I460V mutant showed the best amination activity with a product concentration in the reaction system of 49.1% (Fig. 3b). Contrary to our initial expectations, the double mutant L256V-I460V showed much better activity than the triple mutant L256V-I460V-E484D (product concentrations in the reaction system: 7.2%), indicating that residues at positions 256 and 460 were the most crucial mutation sites. Additionally, it is intriguing to note that when the E484D mutation was combined with other mutations, it consistently led to a decrease in the activity of the original mutants, which indicated that the multi-point mutation combined with E484D seems to be unfavorable to ensure a correct binding position of β-MeCA in the active site pocket.

To further evaluate the catalytic efficiency of the PcPAL-L256V-I460V mutant, we measured its kinetic constants in the deamination reaction using (2S, 3R)-β-MePhe as substrate (Fig. S6). The results revealed a relatively high Km value (Km = 0.7304 ± 0.1195 mM), which indicated the low affinity of (2S, 3R)-β-MePhe to the mutant. This weak affinity allows the product to leave the catalytic pocket more easily during the amination reaction, thereby facilitating the amination reaction.

## Determination of the absolute configuration of the amination product

Encouraged by the above findings, we elucidated the absolute configuration of the amination products by employing two distinct methods. In the first method, by utilizing the three β-methyl phenylalanine (β-MePhe) standards with 2S,3S-, 2S,3R-, 2R,3R- configurations, the in vitro deamination reactions were conducted using the most efficient variant (PcPAL-L256V-I460V). High-performance liquid chromatography (HPLC) analysis demonstrated that only (2S,3R)-β-MePhe was deaminated with full (~100%) conversion (Fig. 4a). Given the reversible nature of PAL-mediated reactions, we deduced that the stereo configuration of the amination product should hence be 2S,3R. To further validate our hypothesis, we conducted the second approach, in which Marfey's reagent was used to derivatize the above three β-MePhe standards as well as the crude amination reaction solution which was formed by the PcPAL-L256V-I460V mutant. As (2R, 3S)-β-MePhe was not commercially available, the β-MePhe mixture standard which contains all four configurations was chosen for the derivatization reaction. After analyzing this by LC-HRMS, we detected the derivatization product of (2R, 3S)-β-MePhe (blue peak in Fig. 4b viii), confirming the presence of the (2R, 3S)-typed substrate in the standard mixture. Moreover, we found that when the crude amination reaction solution was reacted with Marfey's reagent, a peak with the same retention time was observed as the one in the (2S, 3R)-type standard (Fig. 4b and Fig. S7). Hence, the results from both experiments clearly confirmed that the absolute configuration of the amination product was 2S, 3R. Furthermore, the amination reaction catalyzed by the PcPAL-L256V-I460V mutant proceeded with excellent diastereoselectivity (dr > 20:1) and enantioselectivity (ee > 99.5%).

## Development of a whole-cell biotransformation process

To enhance the practicality of this direct asymmetric amination reaction for synthesizing β-branched aromatic amino acids, we developed a whole-cell biotransformation (Fig. 1c)[44,56]. Therefore, the gene of the PcPAL-L256V-I460V mutant was introduced into the strain *E. coli* BL21 (DE3) ΔtyrB, which lacks the aromatic amino acid transaminase gene tyrB and thus avoids potential transamination of the generated product within the cells[58]. After expressing the PcPAL-L256V-I460V mutant at 28 °C for 20 h, the cells were harvested and then resuspended in ammonia buffer for the biotransformation. To make this process efficient, we optimized different conditions by using β-MeCA as substrate and a commonly used 5 M ammonium hydroxide buffer (NH₄OH buffer, pH 10) as the source of ammonia[48,50]. Considering the possible membrane permeability issues, clarified lysate of the harvested cells was used for transformation. However, the results showed that employing whole-cells containing the PcPAL-L256V-I460V gave significantly higher conversions than using clarified lysate, presumably due to higher enzyme stability in vivo under the same catalytic conditions (Fig. S8). Moreover, by resuspending the cultivated cells in NH₄OH buffer to reach an $OD_{600}$ of 30, four distinct substrate concentrations (0.5, 1.0, 1.5, and 2.0 g/L) were systematically investigated. The maximum product concentrations in the reaction system (71%) was achieved at a substrate concentration of 1.0 g/L after 18 h reaction time. Longer reaction times lead to a gradual decline indicating a potential shift in the reaction equilibrium towards the reverse deamination reaction (Fig. 3c). Optimal reaction conditions were thus identified using 5 M NH₄OH buffer, 1 g/L substrate concentration, and 18 h reaction time.

## Exploration of the substrate scope

Given the excellent stereoselectivity demonstrated by PcPAL-L256V-I460V, as well as the simple and practical whole-cell biotransformation method, we next explored the substrate scope of this amination reaction. 15 β-MeCA analogs with different substituents at the phenyl ring were firstly synthesized using a straightforward chemical method (Fig. S9)[59]. Considering that the amino acids around the phenyl ring of substrates in the active pocket of PcPAL, such as L134, F138, and F138, could also potentially influence the conformations of substrates, we conducted screening assays on the synthesized substrates using our mutant library. The HPLC results showed that six substrates (**1, 2, 4** to **8**) could be converted by the PcPAL-L256V-I460V mutant. Despite substrate **3, 9,** and **10** could be converted by both PcPAL-F137V-I460V and PcPAL-L256V mutants, the combination mutant PcPAL-F137V-L256V-I460V showed higher conversion for these three substrates (Fig. S10), highlighting the crucial role of the L256 position in enhancing reaction activity.

Next, we scaled up the reactions using the *E. coli* based whole-cell biotransformation method by using 30 mg substrates and obtained the ten β-substituted amino acid products with yields ranging from 41% to 71% (Fig. 5). In previous reports, the absolute configuration of β-substituted amino acid products were typically determined by referencing to previously reported NMR and specific rotation data[24]. For the newly synthesized β-substituted amino acid products reported here, we assigned the absolute configurations for four products (**4a, 5a, 8a,** and **9a**, Fig. 5b, Table S2 to S5) by X-ray crystallography and this unequivocally confirmed that the absolute configuration of all four crystallized products were 2*S*,3*R*, in accordance with the absolute configuration determined for β-MePhe **1a** (Fig. 4). In addition, a comprehensive analysis of all the products was carried out, including NMR and specific rotation data analysis. Remarkably, all these analytical data consistently confirmed that the configurations of the produced β-branched phenylalanine analogs were 2*S*, 3*R*. All chiral reaction products were obtained with high diastereoselectivity (dr > 20:1) and optical purity (ee > 99.5%) (Fig. 5).

Next, by comparing the yields of these productive reactions, we identified three factors that could affect the amination reaction efficiency. Firstly, the position of substitutions plays a significant role. When the electronegativity of the substituents was the same, substrates with *para* substitutions (e.g., **3a, 4a, 5a,** and **8a**) exhibited higher reactivity compared to those with *meta* substitutions (e.g., *p7*). Notably, substrates with *ortho* substitutions, such as **11** and **12** (Fig. S9), were not converted. Secondly, the electronegativity of substituents could also influence the outcomes. A comparison of yields between products **3a**–**5a** and **6a**, as well as **7a** and **10a**, indicates that electron-withdrawing groups could benefit the conversion. Thirdly, the size of the substitutions at the β-position also plays a crucial role. Although mutants could accommodate the CF₃ group at the β-position, which is similar in size to the methyl group, the yield of **2a** remained relatively low (41%). The introduction of this strong electron-withdrawing CF₃ substituent should reduce the nucleophilicity of the α carbon, thus resulting in a slow amination process. We thought that this accounts for the diminished activity observed for the CF₃-substituted substrate. However, the ability of mutants to recognize the CF₃-substituted substrates strongly implies that steric hindrance also plays a central role in impeding the reaction. Conversely, when larger substitutions like ethyl, propyl, and isopropyl were present at the β-position (**13** to **15**, Fig. S9), no reaction was observed. This observation further underscores the high sensitivity of PcPALs' active sites to steric effects at the β-position of the substrates.

## Elucidating the molecular basis of PcPAL-L256V-I460V for converting β-MeCA

To reveal the molecular basis for accepting β-MeCA, as well as the excellent stereoselectivity during the PcPAL-L256V-I460V mediated amination reactions, the protein structure of this mutant was predicted using AlphaFold 2 and docked with β-MeCA into its active pocket. Subsequently, a series of computational simulations were conducted based on the docking results. Interestingly, similar simulation results were observed in the (β-MeCA)/(PcPAL-L256V-I460V) complex compared to the (CA)/(PcPAL-WT) complex. Specifically, in both complex systems, angles conforming to the Bürgi-Dunitz angle theory were detected, and both angles were around 109° (Figs. 6a and 2a). In addition, in the (β-MeCA)/(PcPAL-L256V-I460V) complex, the mean pocket volume of β-MeCA was around 207.9 Å³, much larger than that in the (β-MeCA)/(PcPAL-WT) complex (177.4 Å³, Figs. 6b and 2d), meaning that β-MeCA was more flexible in the mutant. Furthermore, the optimized conformation of the (β-MeCA)/(PcPAL-L256V-I460V) complex was determined through QM/MM simulations, revealing a distance of 3.28 Å between the hydroxyl hydrogen of Tyr110 and the carboxyl group of β-MeCA. This distance is greater than that from the hydroxyl hydrogen of Tyr110 to βC (2.11 Å) (Fig. 6c), with a corresponding hydrogen bond angle of 118.63° (Table S6, Source Data 5), indicating that the strong hydrogen bond interaction present in the (β-MeCA)/(PcPAL-WT) complex (Fig. 2f) does not exist in the mutant. Overall, the simulation results shown in Fig. 6a–c demonstrate that the conformation of β-MeCA in the mutant was almost same to that of CA in PcPAL-WT. Considering the conformation of CA in the PcPAL-WT, this could be used as a reference for assessing the potential occurrence of reactions when β-MeCA is used as a substrate. Indeed, the simulation results indicated that the positioning dynamic of β-MeCA was relatively flexible within PcPAL-L256V-I460V mutant, which could facilitate the amination reaction process.

To gain a deeper understanding of why β-MeCA showed different positioning dynamics within the PcPAL-WT and the PcPAL-L256V-I460V mutants, we further analyzed another two distances (distance 3: from βC of β-MeCA to amino acid 256; distance 4: from the benzene ring center of substrate to amino acid 460 in Fig. S4). In the complexes (β-MeCA)/(PcPAL-L256V-I460V) and (CA)/(PcPAL-WT), two distinct

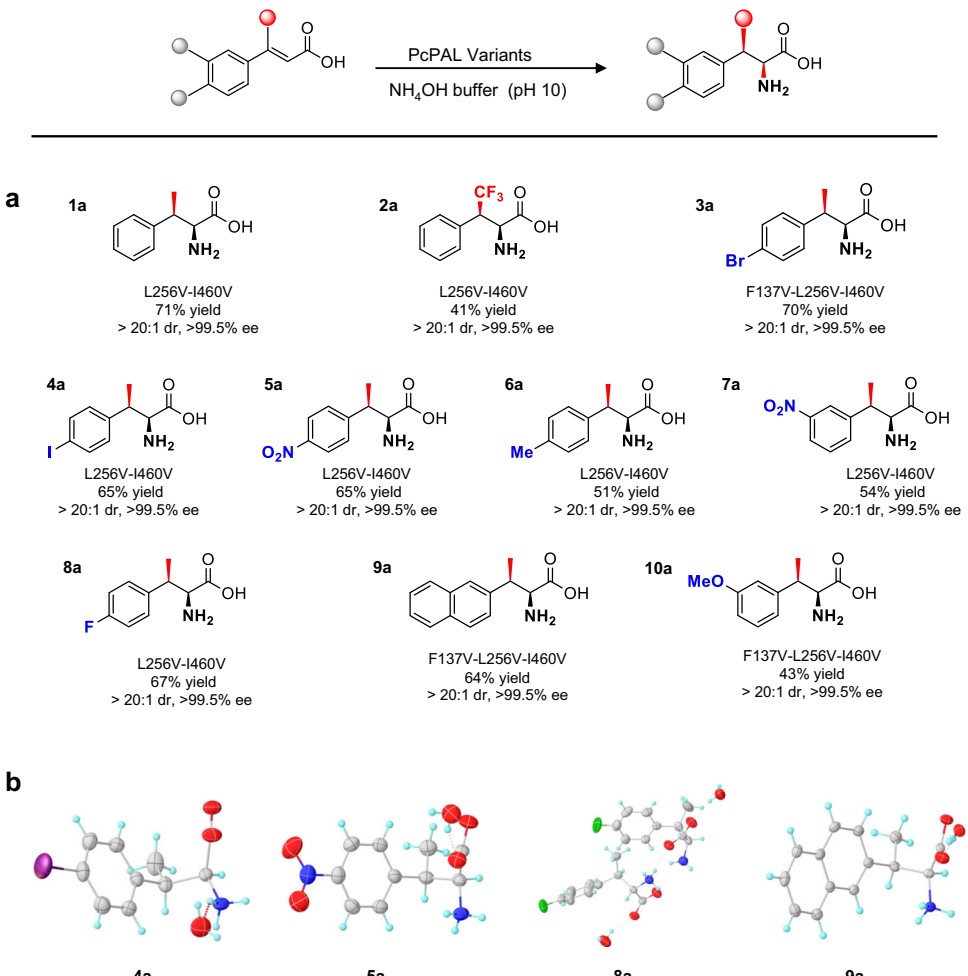

**Fig. 5 | Substrate scope exploration and X-ray crystal structures of the amino acids formed. a** Structures of the produced β-branched aromatic α-amino acids. Yields refer to the isolated product after purification. **b** X-ray crystal structures of products **4a, 5a, 8a** and **9a**. Different colors represent different atoms: gray for carbon atoms, cyan for hydrogen atoms, red for oxygen atoms, blue for nitrogen atoms, and green for fluorine atoms. Source data are provided in Source Data 5.

low-energy regions can be observed, each characterized by a broader primary energy well separated by a lower energy barrier (-1.2 kcal/mol), indicating that the substrate can readily traverse this barrier to adjust its conformation (Fig. 6d, e). Within the main low-energy regions, in the (β-MeCA)/(PcPAL-L256V-I460V) complex, distance 3 ranges from 7.9 to 8.6 Å, and distance 4 ranges from 8.0 to 8.9 Å (Fig. 6d). Similarly, in the (CA)/(PcPAL-WT) complex, distance 3 ranges from 7.4 to 8.9 Å, and distance 4 ranges from 8.3 to 9.8 Å (Fig. 6e). However, in the (β-MeCA)/(PcPAL-WT) complex, only one low-energy region is observed, with distance 3 ranging from 7.9 to 8.6 Å, and distance 4 ranging from 8.0 to 8.9 Å (Fig. 6f). These results indicate that compared to the (β-MeCA)/(PcPAL-WT) complex, β-MeCA in (PcPAL-L256V-I460V) exhibits a greater range of motion and a broader conformational space. Considering the wider angular range between MIO's N atom and the double bond in (β-MeCA)/(PcPAL-L256V-I460V) and (CA)/(PcPAL-WT) complexes, these findings support our hypothesis that substrate positioning dynamics influence the occurrence of the reaction.

Additionally, the unique and precise distribution of amino acids in the PcPAL's substrate binding pocket determines the stereoselectivity of the asymmetric amination reaction. The amino acids around the benzene ring are mainly hydrophobic, while the ones around the carboxyl group are more hydrophilic (Fig. 6c). Guided by the unique distribution of amino acids within the pocket, the β-methyl group of β-MeCA demonstrates a preferential orientation towards the hydrophobic region. This will be advantageous for the enzyme's adaptation and stereoselectivity towards β-MeCA analogs. Importantly, due to the fact that the critical MIO group is positioned on the *si-si* face of the substrate plane, the chiral center formed after the amino group attacking αC can only be 2*S*. Additionally, the crucial Tyr110 is located on the *Re-Re* face of the substrate plane. In such case, when the βC is protonated, the chiral center formed can only be 3*R*. Overall, the distinctive amino acid distribution provides essential molecular foundations for correctly accommodating β-MeCA and facilitating a highly stereoselective asymmetric amination reaction.

## Discussion

In summary, through computational analysis and a rational protein engineering approach, we successfully elucidated and addressed the long-standing challenge of PAL's inability to aminate β-substituted CA analogs. Our method for the direct asymmetric amination of β-substituted CA analogs to generate β-branched aromatic α-amino acids possesses several advantages, such as readily available substrates, excellent atom economy, no need for cofactors, and high regio- and stereo selectivity. These results represent a significant advancement in the field of PALs, contributing not only to the expansion of their applications but also providing an approach for the synthesis of β-branched aromatic α-amino acids.

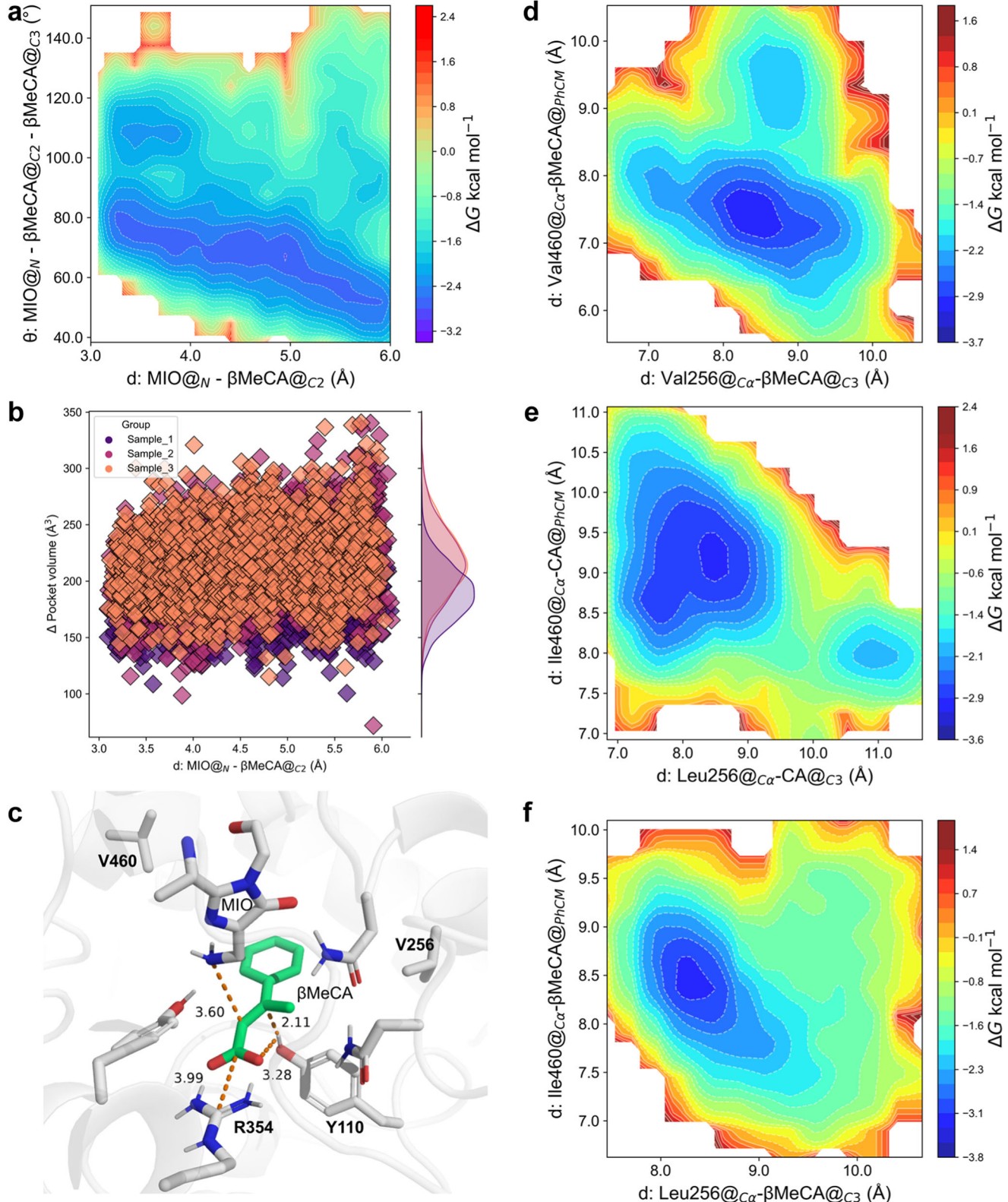

**Fig. 6 | Molecular basis explaining the amination of the β-MeCA substrate by the mutants. a** Gibbs energy diagram of the cMD simulations projected onto the distance from the amino N of MIO to the αC of the substrates and the angle between MIO and the double bond of the substrates within the (β-MeCA)/(PcPAL-L256V-I460V) mutant. **b** the pocket volume detected during the US simulations of the (β-MeCA)/(PcPAL-L256V-I460V) complex. **c** Optimized pre-reaction states of β-MeCA in the PcPAL-L256V-I460V mutant using the Quantum Mechanics/Molecular

Mechanics (QM/MM) methodology. **d, e, f** Gibbs energy diagram of the cMD simulations projected at the distance from βC of β-MeCA to amino acid 256 and the benzene ring center and amino acid 460 within the (β-MeCA)/(PcPAL-L256V-I460V) complex, (CA)/(PcPAL-WT) complex, and (β-MeCA)/(PcPAL-WT) complex respectively. Source data are provided in Source Data 3 to Source Data 5. CA cinnamic acid, β-MeCA β-methyl cinnamic acid, MIO 3,5-dihydro-5-methylidene-4H-imidazol-4-one.

## Methods

### Construction of PcPAL mutant libraries

The plasmid PcPAL-pETs[58] was used as template and site-directed mutagenesis using the QuikChange® method was used for the creation of mutants. Primers were designed using the Agilent (USA) web tool QuikChange® Primer Design and provided in Table S1. The PCR protocol employed the Pfu Plus polymerase from Roboklon (Germany) using the following steps: initial denaturation at 98 °C for 30 s, followed by 25 cycles of denaturation at 98 °C for 10 s, annealing at 68 °C for 10 s, and extension at 72 °C for 3.5 min. A final extension step was performed at 72 °C for 10 min. The PCR products were then digested using DpnI and used for the transformation of *E. coli* (Top 10) chemical competent cells through heat shock. The introduced mutations were confirmed by sequencing.

### Cloning, expression and purification of PcPAL and associated proteins

The plasmids harboring the genes encoding the PcPAL variants were inserted into the *E. coli* BL21(DE3) strain by heat shock transformation. The transformed cells were incubated overnight in a culture tube containing 4 mL LB medium supplemented with kanamycin (40 mg/mL) at 37 °C with constant shaking at 180 rpm. The overnight cultures were then transferred to a 500 mL flask containing 100 mL of TB medium supplemented with kanamycin (50 mg/mL) and incubated at 37 °C with constant shaking at 180 rpm. Once the cultures reached an $OD_{600}$ of 0.6, the temperature was lowered to 28 °C, and the cultures were incubated for an additional 20 h with constant shaking at 180 rpm. Cells were centrifuged at $10,000 \times g$ for 30 min, and the cell pellet was resuspended in 20 mL lysis buffer (25 mM HEPES, pH 7.5, 300 mM NaCl, 5 mM imidazole, 10% glycerol) and lysed using ultrasonication. The insoluble debris was removed by centrifugation at $10,000 \times g$ and 4 °C for 1 h. The protein supernatant was incubated with 1 mL of Ni-NTA Sepharose resin for 0.5 h with slow and constant rotation at 4 °C. The protein-resin mixtures were then loaded onto a gravity flow column, and the proteins were eluted using increasing concentrations of imidazole (25 mM, 50 mM, 300 mM) in Buffer A (25 mM HEPES, pH = 7.5, 300 mM NaCl, 10% glycerol). The purified proteins were further subjected to desalting using PD-10 desalting columns and buffer B (25 mM HEPES, pH = 7.5, 50 mM NaCl, 10% glycerol), followed by concentration using an Amicon Ultra-4 (GE Healthcare) centrifugal filter. The purity of the proteins was assessed using 12% acrylamide SDS-PAGE (Fig. S11).

### General method for screening in vitro amination reactions

A purified PcPAL variant (100 μg) was added to a solution of the substrate (1 mM, 500 μL) in NH$_4$OH buffer (5 M, pH = 10, adjust by H$_2$SO$_4$). The mixture was incubated at 30 °C, 500 rpm for 24 h. After that, the reaction was stopped with 500 μL 10% H$_2$SO$_4$ (v/v), thoroughly shaken and centrifuged ($12,000 \times g$, 5 min). The supernatant was transferred to a filter vial and used directly for UPLC analysis. UPLC analysis of reactions were performed on a Luna® Omega column (C18, 5 μm, 150 × 4.6 mm, Phenomenex) at a flow rate of 1 mL min$^{-1}$ and detected at 210 nm over a 10 min gradient program with water (eluent A) and acetonitrile (eluent B): $T = 0$ min, 5% B; $T = 4$ min, 60% B; $T = 6$ min, 5% B; $T = 10$ min, 5% B.

### Biotransformations and product purification

Recombinant strains were inoculated in TB medium (2 L) by growth to an $OD_{600}$ of 0.8–1.0 at 37 °C. After expression at 28 °C and 220 rpm for 20 h, the cultures were harvested by centrifugation at $4000 \times g$ at 4 °C and washed twice with ddH$_2$O. After that, the cells were resuspended in 30 mL 5 M NH$_4$OH buffer (pH = 10) to reach an $OD_{600}$ of 30, into which different substrates (1 g/L) and 1% Triton X-100 were added. After 18 h incubation at 28 °C, the reaction mixture was acidified slowly with aq. H$_2$SO$_4$ (50% v/v) at 0 °C, centrifuged ($8000 g$, 10 min, 4 °C) and applied

to a column containing washed Dowex 50 W resin for desalination. The resin was washed with water until the eluate was neutral, then the product was eluted with aq. NH$_4$OH (15% v/v). The fractions containing the product were pooled and concentrated by evaporating in a centrifugal evaporator. After that, the crude products were further separated by semi-preparative HPLC (SHIMADZU, SPD-20A 230 V) equipped with a Phenomenex Luna C18 reverse phase column (5 μM, 250 × 10 mm). The purification process involved isocratic elution with a mixture of 12% acetonitrile and water containing 0.1% formic acid for a duration of 30 min.

### NMR spectroscopy

The NMR spectra were recorded on a Bruker Avance III spectrometer at a $^1$H frequency of 300 MHz or 400 MHz or 600 MHz. Samples (varying from 1 to 15 mg) were dissolved in 400 μL MeOH-$d4$ or D$_2$O (Cambridge Isotope) and all spectra were recorded at 25 °C (298 K). NMR experiments were processed with the Bruker Topspin program and analyzed with the MestReNova software (Version 6.1.0-6224).

### X-ray crystallographic analysis

Compounds **4a**, **5a**, **8a**, and **9a** were obtained as colorless crystals by recrystallization from a water/methanol system using the vapor diffusion method. The single-crystal X-ray diffraction data were collected using the XtaLAB Synergy Custom and XtaLAB PRO MM007HF diffractometers with Cu-Kα radiation ($\lambda = 1.542$ Å). The structures were solved by the intrinsic phasing solution method. The models were refined with ShelXL-2018/3 using least-squares minimization. The structures were visualized by Olex2 (version 1.5).

### Preparation of the MIO-group containing tetramer structures of PcPAL and docking analysis

AlphaFold 2 was used to predict the tetrameric form 3D structures of PcPAL and its related mutants. The MIO structure within them was derived from a resolved crystal structure. We extracted the MIO-group from the crystal structure (PDBID: 6RGS) and re-edited it in PyMOL (version 1.8.0.2) to aminate the MIO-group. Then this non-standard amino acid was checked and capped, and structural optimization and vibrational frequency checks were conducted using ORCA 5.0.4[60] at the r2SCAN-3c[61] theory level, using the optimized conformation in CPCM ($\varepsilon = 4$) implicit solvent as the initial conformation for MIO to ensure the accuracy of its geometry and electronic structure. Subsequently, RESP2 charge fitting was carried out using Multiwfn_3.8_dev[62], and it was parameterized using the Amber/GAFF2 atom types through the Antechamber tool. Finally, the missing parameters were completed using parmchk2, and the results were visually inspected and subjected to dynamic testing. After that, the docking experiments were carried out on Autodock Tools (version 1.5.6), with the grid size restricted to a $10 \times 10 \times 10$ Å box centered on the midpoint between the MIO residue and Tyr110 of the tyrosine loop. PyMOL (version 1.8.0.2) was used to visualize 3D structure analysis and comparison.

### Conventional molecular dynamics (cMD)

Protonation prediction for the protein was performed using H++[63], and the initial structure underwent 2,500 steps of steepest descent and 2,500 steps of conjugate gradient optimization using the Amber22 program[64]. The protein residues were parameterized using the AMBER14SB force field. The system was solvated using a periodic dodecahedron with a 10 Å buffer between solute atoms and the box edges, and appropriate amounts of Na$^+$ and Cl$^-$ ions were added to neutralize the system. Subsequently, a 2 ns NVT heating simulation was conducted to equilibrate the molecular system. The NVT simulation employed the Langevin thermostat to gradually raise the temperature to 300 K, with hydrogen bonds constrained using the SHAKE algorithm and a time step of 2.0 fs. This was followed by a 5 ns NPT simulation to further relax the system, also using the Langevin thermostat

to maintain the temperature at 300 K, with hydrogen bonds constrained using SHAKE and a time step of 2.0 fs. Long-range electrostatics were calculated using the particle mesh Ewald method with a non-bonded interaction cutoff of 10 Å. The MD simulations were run for 100 ns with a 2 fs time step, saving coordinates every 2 ps to the trajectory file. The simulation trajectories were analyzed using the cpptraj tool.

## Umbrella sampling simulations

The initial coordinates for umbrella sampling were derived from equilibrated conformations obtained during the MD phase, followed by three independent 10 ns NPT simulations to serve as the starting points for the three parallel sets of umbrella sampling. Umbrella sampling (US) utilizes a reaction coordinate describing the distance between two heavy atoms involved in the nucleophilic addition reaction (the N atom of MIO to the αC of CA or β-MeCA), aiming to reveal differences between the WT and mutant in the pre-reactive state. We employed 38 simulation windows with a step size of 0.75 Å between windows. The system was constrained to other values by introducing an artificial harmonic bias potential with a constraint force of 50 kcal/mol. Each sampling window underwent 5,000 steps of energy minimization and 50 ps of pre-equilibration, followed by a 2 ns NPT simulation. Throughout the umbrella sampling process, a time step of 1.0 fs was used, and other simulation parameters were consistent with pre-equilibration. Three repetitions were carried out starting from different equilibrated points. Weighted Histogram Analysis Method[65] was employed to compute the potential of mean force (PMF).

## Analysis of the USMD results

Cpptraj and PLUMED[66] were used to analyze the umbrella sampling trajectories to extract atomic distance and angle information. The protein pocket volumes and substrate volumes during umbrella sampling were calculated using Fpocket (https://github.com/Discngine/fpocket, Accessed on June 2, 2024) and MoloVol[67], respectively. For protein pocket volume calculation, the min_alpha_size was set to 2 Å, and the max_alpha_size was set to the default value. For molecular volume calculation, the rprobe was set to 1.2 Å, and the grid was set to 0.1 Å. The volume difference calculation formula is as follows:

$$\Delta V = V_{protein} - V_{substrate}$$

After obtaining the results of the umbrella sampling, the coordinates of the corresponding window at the lowest energy reaction coordinate were read in. A 2 ns classical molecular dynamics (cMD) simulation was performed to sample the pre-reaction state. The structures from the pre-reaction state sampling were clustered using cpptraj, and potential errors in the clustering process were minimized using 2,500 steps of steepest descent and 1,000 steps of conjugate gradient minimization.

In the pre-reaction state sampling, we first read the coordinates corresponding to the lowest energy point from the PMF curve obtained through umbrella sampling. Then, we performed a 2 ns cMD simulation starting from these coordinates, recording snapshots every 1 ps. Finally, we used cpptraj to perform clustering analysis on these snapshots, setting a total of 5 conformational clusters.

## Quantum mechanics/molecular mechanics (QM/MM)

Based on clustering results, we optimized the conformations using QM/MM. This approach allows us to incorporate electronic correlation effects within the system at a higher quantum mechanics (QM) level, while describing interactions using molecular mechanics (MM) methods, aiming to achieve more accurate and reliable interaction descriptions. The QM/MM interface was executed using the ASH program (https://github.com/RagnarB83/ash, Accessed on June 10, 2024.). The QM region including the substrate, MIO, ARG354, TYR110,

TYR351, ASN260, and ASN384, was treated with the BLYP function[68], def2-SVP basis set[69], def2/J auxiliary basis set[70], and D3BJ dispersion correction[71] level using ORCA 4.0.5. The MM region was described using OpenMM 8.0[72] and Amber ff14SB force field. To handle electrostatic interactions between QM and MM layers, the electrostatic embedding technique was applied, and the link atom method resolved boundary issues between QM and MM layers. The active region is defined as within a 10 Å radius around the amino N of MIO. This method allows us to simulate the interactions of the wild-type and mutant in the pre-reaction state more precisely.

## Determination of kinetic constants

The kinetic constants were measured in triplicate at 30 °C using 96-well UV-microplates. The reactions were carried out in 50 mM PBS buffer (pH 8.0) and the mutant enzyme (PcPAL-L256V-I460V) was maintained at a constant concentration of 0.6 μM, with eight different substrate concentrations ranging from 0.015625 to 1.5 mM. Kinetic measurements were performed using UV spectroscopy to monitor the production of β-MeCA over a 10 min period at a wavelength of 260 nm. The kinetic constants were then calculated from Michaelis-Menten plots via non-linear regression analysis.

## Reporting summary

Further information on research design is available in the Nature Portfolio Reporting Summary linked to this article.

## Data availability

Crystallographic data for the structures reported in this article have been deposited at the Cambridge Crystallographic Data Centre, under deposition nos. CCDC 2293061 (**4a**), 2293062 (**8a**), 2293063 (**5a**) and 2313111 (**9a**). Copies of the data can be obtained free of charge via https://www.ccdc.cam.ac.uk/structures/. The crystal structure of PcPAL (PDB: 6HQF) is available at https://www.rcsb.org/structure/6HQF. All other characterization data and detailed experimental procedures are available in the supplementary information file. Source data are provided together with this publication.

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

## Acknowledgements

We thank Prof. Dr. Andreas Link and Maria Hühr for access to the 400 MHz NMR spectrometer and Ina Menyes for her support with the HPLC analytics. This work was supported by the European Union's Horizon 2020 Research and Innovation Program under Grant Agreement No. 814650 for the project SynBio4Flav, and supported by the start funding of Huazhong University of Science and Technology to Guifa Zhai (3034514105). We thank Siwen Yuan and Feng Li for specific rotation data collection.

## Author contributions

C. Sun, G. Zhai, and U.T. Bornscheuer designed the project. C. Sun, G. Zhai, G. Lu, B. Chen, G. Li and Y. Brack performed the experiment; C. Sun, G. Zhai, G. Lu, Y. Wu, G. Li, B. Chen, Y. Brack, D. Yi, Y. Ao, S. Wu, R. Wei, Y. Sun and U.T. Bornscheuer analyzed and discussed the results. C. Sun, G. Zhai, G. Lu and U.T. Bornscheuer wrote the manuscript with inputs from all authors.

## Funding

## Competing interests

The authors declare no competing interests.
