## [Peer Review File · Nature Communications]

Direct Asymmetric Synthesis of β -Branched Aromatic α -Amino Acids using Engineered Phenylalanine Ammonia LyasesREVIEWER COMMENTS

Reviewer #1 (Remarks to the Author):

This manuscript by Sun, Zhai, Bornscheuer and colleagues describes a new biocatalytic method for the asymmetric amination of β -substituted cinnamic acids to generate β -branched aromatic α -amino acids through the utilization of engineered phenylalanine ammonia lyases (PAL). In this work, the authors addressed the long-standing challenge of PAL biocatalysis in the amination of β -substituted cinnamic acids by using computational analysis and a rational protein engineering. This reaction showed good substrate scope and afforded the β -branched phenylalanine analog products in excellent diastereoselectivity (dr up to > 20:1) and enantioselectivity (ee up to 99.5%). Moreover, this reaction can also be scaled-up using whole-cell biotransformation, underscoring its potential for further application. Overall, this manuscript represents a valuable advance in the field of PAL biocatalysis and the synthesis of non-proteinogenic amino acids using enzymes. Additionally, the manuscript and supporting information are prepared with good quality. In summary, publication of this manuscript in Nature

Communications is recommended if the authors can address the following issues.

- 1) Does the Z/E configuration of the β -MeCA substrate influence the final diastereoselectivity? For example, what result will be obtained when employing (Z)-1 rather than (E)-1 in the standard reaction conditions?
- 2) The yield for the standard reaction decreased after 18 h and the time curve of substrate concentration showed this might result from the reverse deamination reaction (Fig. 3c). Rationales are needed to account for the fact that these reactions did not reach equilibrium after a long period of reaction time. Is it due to the concentration change of the NH₃ source in the reaction?
- 3) As mentioned in question 2, the reaction time will influence the final yield. For some tested substrates that showed lower yields (2a, 6a, 10a in Fig. 5a), what results will be

attained after

either prolonging or shortening the reaction time for these specific substrates?

4) For supplementary materials, the ^{13}C NMR spectra shown in Fig. S6/S11/S22/S26 are unsatisfactory. Please pay attention to the ^{13}C - ^{19}F coupling in fluorine-containing compounds such as substrates 2/8, products 2a/8a. Proper description of chemical shifts and coupling constants is needed in the data characterization section.

Reviewer #2 (Remarks to the Author):

General comments:

The manuscript describes the extension of the substrate scope of phenylalanine ammonia lyases (PAL) for the asymmetric synthesis of β -branched aromatic α -amino acids. By the aid of computational analyses, the possible reasons behind the PAL's inability to accept β -methyl cinnamic acid (β -MeCA) as substrate have been delivered, while using a double (PcPAL-L256V-I460V) and a triple mutant (PcPAL-F137V-L256V-I460V) the successful synthesis of several β -branched phenylalanines have been performed.

The result is of high significance to the field of biocatalysis, providing the first successful application of PALs in the synthesis of phenylalanines with substituent in the β -position and the proposed molecular mechanism for the inability of the wild-type PAL to transform these β -branched substrates. Accordingly, while the manuscript employs mutant variants of the hydrophobic binding pocket of PcPAL, recently explored and mapped by mutational and structural analysis, the application of PcPAL mutants to transform the β -branched substrates is highly original in respect with the state of art. The interaction of the catalytically essential Tyr110 with the carboxyl group of the substrate, instead of assisting at the β -deprotonation, is also highly original. However in this later result, some scepticism should be formulated: considering the i) high flexibility of the Tyr110-loop, supported by its different positioning/or absence among the reported crystall structures of PcPAL (1W27, 6RGS, 6F6T, 6HQF) and the ii) limited level of accuracy of predicting the positioning of flexible loops by computational approaches, the molecular level rationale based solely on computationally assessed atomic distances is considered a weak point of the manuscript. The experimental structure (e.g. of the inactive wt-PcPAL with the β -Me-Phe) confirming the estimated distances would greatly

improve the manuscript's impact and bring the ultimate support for the described molecular level mechanism.

While the methodology is sound, in several cases the results interpretation, methodology development and description must be improved that the work to be reproduced.

In this context major clarifications/improvements should be provided for a novel round of evaluation.

Major issues:

1. Introduction part, paragraph starting with : "In previous reports, PALs from.... " - lit 36-40 are not about engineering these PALs, as stated by the phrase, only two of cited references deal with PAL engineering. Lit 33-36 is not exhaustive, is only selection from the literature dealing with these PALs, and they apparently do not share a common selection criteria (e.g. papers dealing with their substrate scope description or papers reporting their first discovery). At this part, references and their selection should be improved.

2. Related to Figure 2c and 2d:

- Fig 2d should present the substrate in the catalytic site at the same size as Figure 2c, the cartoon background makes it difficult to follow the details

- please label important residues which are mentioned in the Results section, part including the discussions related to Fig. 2c and Fig. 2d.

The same is valid for Fig. 6a

Residue R354 should also appear as reference for its correct positioning for the fixation of the substrate's carboxyl group (including in Fig. 3 and Extended data 1.) The distances related to this residues should also be monitored, presented within the selected docking results.

3. From Figure 3a, L206 does not seem to locate near the β -methyl substituent of the substrate, also the data from literature (lit 44 from the references) argue that this residue is in close proximity to the ortho-position of the substrate's phenyl ring.

In Figures with the catalytic site the R354 should also appear as reference for its correct positioning for the fixation of the substrate's carboxyl group.

4. Section Rational engineering of PcPAL for the amination of β -MeCA, phrase: Additionally, we also investigated three other mutants, previously established in our laboratory (E484D, F137V-I460V40, and F137A-I460A40) to explore their potential.

- please mention the origin and the previous utility of double mutant F13V-I460V/A

5. Figure 4 - Provide in ESI the original chromatograms.

Legend of Figure 4a - What is the difference between v and v_i ? - in each case deamination of (2R,3R)- β -MePhe is written in the figure caption

6. Section "Development of a whole-cell biotransformation process":

a) Why 5M ammonia and 4 M ammonium carbamate buffer was tested?

In the PcPAL-mediated reactions other optimal ammonia concentrations are reported, such as 2 M ammonium carbamate, 4 M NH₄OH. Further ammonium carbamate might release 2 eq. of ammonia, thus it would be equivalent to 8 M NH₄OH, in this way it is hard to compare it with the 5M NH₄OH buffer. Please provide some rationale for the employed ammonia concentrations.

b) "... we found that 5 M ammonium hydroxide buffer (NH₄OH buffer, pH 10) performed better than 4 M ammonium carbamate buffer (NH₂CO₃NH₄ buffer, pH 10) (Fig. S4)." and later: "...gave significantly higher conversions than using clarified lysate, presumably due to higher enzyme stability in vivo under the same catalytic conditions (Fig. S4)." - please include somewhere the conversion values related to chromatograms from Fig.S4.

7. Section "Elucidating the molecular basis of PcPAL-L256V-I460V for catalyzing β -MeCA"

It is stated that the protein structure of the mutant variant was predicted by AlphaFold. The build-up of the MIO-group how was achieved by using AlphaFold? And Tyr110-loop? Please comment and provide details in the ESI related to this model built-up, since according to the reviewers experience, the MIO-group cannot be built up by Alpha-Fold and also the Tyr-loop is modelled too tightly or loosely within the catalytic site.

8. Kinetic data for the transformation of β -MeCA or the reverse reaction catalyzed by the PcPAL-L256V-I460V should be also assessed, in order to position within the state of art the catalytic efficiency of the mutant variants towards these substrates.

Data, methods within supporting information should be seriously improved in terms of quality:

9. Supporting information Section 1.1. - Reference 2 is too general to be acceptable, please replace with more specific references or provide page-numbers from this book.

10. Supporting information, section Optimization of the in vivo amination reactions

"...These reactions were placed at 250 rpm and 30 °C for 24 h. Then 100 μ L samples were extracted and supplemented with 100 μ L aq. H₂SO₄ (10% v/v) to stop the reaction.

Subsequent centrifugation (12,000 × g, 4 °C, 2 min) separated the supernatant, which was then subjected to filtration and directly injected into an HPLC apparatus for analysis.” Stopping the reaction with acidification, followed by filtration might lead to serious errors in the conversion values. The cinnamic acid substrates under acidic conditions have low water solubility, might form micro suspensions, which together with cell debris are partially removed by filtration. This issue was already observed and reported during the optimization of the work-up procedure of the whole-cell PAL biotransformation within reference 46. This might be and was alleviated by stopping the reaction by adding 50% MeOH, followed by filtration (with or without adding acid – in function of the employed HPLC method) and injection to HPLC. Please check if by this later sample preparation method the same conversions are obtained than the actual conversions.

11. Supporting information page 10 – please provide the original and full chromatograms, the region with the peaks might be zoomed as it appears currently, however the original chromatograms are mandatory.

12. Supporting information – in all cases >99.5% ee values are reported, however in several cases, e.g. 5a, 6a, 8a, the intensity of the peaks relatively to the baseline noise is excluding that such high ee values could be determined. The noise derived from the baseline controls the detection limit of the undesired enantiomer/diastereomer, which should be taken into account. In this sense for the accurate ee measurements probably peaks of higher area would be requested in fact in all cases.

13. NMR spectras - Figures should be amplified on the interesting region, and also during data interpretation and in Figures signals must be linked to the molecules protons.

Some ¹³C-NMR measurements need more scans, the signal intensities are very low, eg. – Figure S6, Figure S11.

In figure S15, ¹H-NMR spectra seems to contain besides the specific signals of the molecule several additional signals, please explain or improve compound purity.

14. Figure S1 - standard deviations are needed.

The use of calibration curve based solely on the product’s signal, might provide large standard deviations in the conversion values, due the effect of small daily differences in analyte and analysis conditions on the peak area of individual signals. This is usually alleviated by the use calibration curves based on relative intensities of both relative substrate and product or the use of internal standard. In the absence of these, please

provide information what are the standard deviations of conversion assessments when the same sample is injected in duplicate or triplicate. This might be surprisingly large, also considering the issues mentioned in point 2.

15. Figure S3 – several chromatograms show additional peaks of non-negligible intensities (e.g. b, e, f) of which appearance and nature should be commented or explained.

16. “Figure S32 .Overlap check of specific windows with histograms” – please describe what represents this image, X-axes what represent, etc.

17. The authors should specify the program used for molecular docking with details, such as the grid size, etc. The software mentioned in the SI, is just a GUI. (correct name is autodock tools 1.5.6)

Minor issues:

1. Introduction part, paragraph starting with : “In previous reports, PALs from.... “ insert space between word ‘...variabilis’ and ‘their’

2. pg. 9, Phrase: “Through analyzing the crystal structure of PcPAL (PDB:6HQF), we found that L256 is positioned in close proximity to the β -methyl group of the substrate (Fig. 3a)” - I guess this is based on the analysis of a docking result, with the β -methyl-Phe docked into 6HQF, not of the crystal structure as it might be interpreted from the text.

3. Please correct PcPAL to its correct form PcpAL

4. Figure 3a – please mention error bars are derived from triplicate or duplicate experiments.

5. First sentence of Section “ Development of a whole-cell biotransformation process”, instead of literature 48, 49, more representative would be the recently optimized whole-cell procedures using PcpAL and/or its mutant variants, references 46 and 45 from the manuscript.

6. Supporting information, pg. 11-16– correct the NH₂- group in the structure of the compounds (number in subscript).

7. Figure S31 – highlighted residues and the MIO-group should be labelled

Reviewer #3 (Remarks to the Author):

In this work, Sun et. al targeted the challenge of the asymmetric synthesis of β -branched

aromatic α -amino acids using engineered phenylalanine ammonia lyases (PALs). This biocatalytic outcome is highly significant because it achieves the desired synthesis through direct amination with a high diastereoselectivity, enantioselectivity, and yield without requiring additional cofactor regeneration system. This work also demonstrates a high level of innovation as it integrates computational tools to guide experimental mutation discovery and to elucidate the mechanisms behind catalysis. Below, we listed critical comments regarding the organization, necessity, and scholarly presentation of the computational components. We hope these comments can help improve the quality of the draft to a publication level:

Major issues:

1. As a specialist in computational modeling, I find the “In silico analyses of PcPAL towards β -MeCA” to be not convincing. In this section, the central point of computational investigation is the origin behind which PcPAL can catalyze CA but not β -MeCA, which is indeed an important question. Based on existing theories, reasonable hypotheses behind the lack of activity could be differences in TS stabilization (Houk, Warshal), ground state destabilization (Hershlag), substrate positioning dynamics (Benkovic), binding, and so on. However, it was not clear to me what fundamental hypothesis the authors made when conducting the computational analysis, albeit that substantial efforts have been spent on comparing the geometric/conformational differences between two complexes using docking, classical MD and QM/MM. I suggest the authors to articulate their hypothesis and frame their computational results around it. For example, the substrate positioning dynamics may be hypothesized to control the selectivity. As such, all comparisons on the reaction coordinate distribution will make sense, as long as the authors provide or cite evidence to justify that TS barriers are similar. Instead, the authors may hypothesize TS barrier difference is the key cause, but how to relate the ground state/reactive conformation to TS is another point of justification.
2. At the end of the computational analyses, the authors propose the molecular principle of mutagenesis. Though appearing reasonable, many statements involved in the chain of reasoning appears inconclusive. First, the docking studies are good for providing candidates for MD and QM, but are not quantitative enough (more than “proved insufficient”) to judge selectivity and conformational differences. Second, the umbrella sampling involves limited conformational sampling and can not “strongly indicate” the lack of PcPAL activity toward β -

MeCA. I am sure that authors realize this and made more substantial sampling in the subsequent section, but it does not help remedy the logical issue in umbrella sampling. Third, IGM analysis is typically used to qualitatively show what interactions are important. The conclusion from the IGM analysis is pretty vague. Fourth, the authors mentioned about different hydrogen bonding patterns between two complexes derived from QMMM calculations, which is pretty interesting, but they fail to justify the number of snapshots involved in the calculations, and which one involves lower energies or major conformational cluster, etc. Overall, I would suggest rewrite the computational section to enhance the internal logic.

3. The authors proposed two pieces of engineering principles from their computational studies – they are “alleviating the stringent active pocket of PcPAL to enhance the conformational flexibility of β -MeCA analogs” and “inducing reactive conformations by engineering not only the residues around the β -methyl group of β -MeCA, but also those around its phenyl ring”. Changing active site pocket is a common practice in rational protein engineering, and many of these insights can be inferred merely from cavity analysis or MD trajectory analysis without extensive umbrella sampling and QMMM calculations. As one example, the key conclusion about the H-bond can be obtained by just analyzing the trajectory of the cMD (the distance distribution as a histogram). The authors would want to justify the necessity of using such an extensive set of computational protocols. In particular, why the same type of insight is difficult to gain without QMMM or umbrella sampling.

Minor issues:

1. (line 130) Please describe how are the starting structures for US selected from the cMD. Please include this structure in the SI. (PDB files in .zip and figures for the active conformations)
2. For texts between line 124 and 145 describe the process of discovery but do not support any statements. They are good fit for SI.
3. (line 150) The setup of this 2nd round of US simulation is vague. What is the starting structure? How is the pre-reactive state defined and modeled?
4. I suggest moving most of the IGM analysis to SI and leaving only the insights about the interactions between Tyr110 and beta-MeCA.
5. The details of IGM setup appears missing. The author needs to describe the detailed selection of fragments in the Method section.

6. Line 167, the QM/MM optimization uses only one frame from the MD or US. The authors may consider moving the QM/MM part to SI and just analyzing the trajectories from cMD for distance distribution, unless they would want to afford QMMM optimization on more snapshots.

7. The 2nd part of the computational study from line 363 suffers from the same problem I commented on for the 1st part in comment #10. Please address this accordingly. The mutants help the catalysis by ground state destabilization (i.e.: the reactant state won't form the H-bond).

8. It will be helpful to readers if the authors also compare the performance of WT in Figure 5.

Reviewer #4 (Remarks to the Author):

I co-reviewed this manuscript with one of the reviewers who provided the listed reports.

This is part of the Nature Communications initiative to facilitate training in peer review and to provide appropriate recognition for Early Career Researchers who co-review manuscripts.

Reply to reviewer comments

Manuscript number: NCOMMS-24-15772-T

Title: Direct Asymmetric Amination Enables the Synthesis of β -Branched Aromatic α -Amino Acids using Engineered Phenylalanine Ammonia Lyases

We express our sincere appreciation to the editor and all the reviewers for their insightful comments, which significantly contributed to the enhancement of our manuscript. We carefully considered all the feedback and performed a number of new experiments as requested by the reviewers. The major modifications have been highlighted with a yellow background in the revised manuscript. Additionally, we provide detailed explanations for the key revisions below.

Reviewer 1:

This manuscript by Sun, Zhai, Bornscheuer and colleagues describes a new biocatalytic method for the asymmetric amination of β -substituted cinnamic acids to generate β -branched aromatic α -amino acids through the utilization of engineered phenylalanine ammonia lyases (PAL). In this work, the authors addressed the long-standing challenge of PAL biocatalysis in the amination of β -substituted cinnamic acids by using computational analysis and a rational protein engineering. This reaction showed good substrate scope and afforded the β -branched phenylalanine analog products in excellent diastereoselectivity (dr up to > 20:1) and enantioselectivity (ee up to 99.5%). Moreover, this reaction can also be scaled-up using whole-cell biotransformation, underscoring its potential for further application. Overall, this manuscript represents a valuable advance in the field of PAL biocatalysis and the synthesis of non-proteinogenic amino acids using enzymes. Additionally, the manuscript and supporting information are prepared with good quality. In summary, publication of this manuscript in *Nature Communications* is recommended if the authors can address the following issues.

We are very grateful to reviewer#1 for this very positive evaluation of our manuscript and also for highlighting the importance of our achievements. We are especially grateful that publication of our work in *Nature Communications* has been suggested after revision.

1) Does the Z/E configuration of the β -MeCA substrate influence the final diastereoselectivity? For example, what result will be obtained when employing (Z)-1 rather than (E)-1 in the standard reaction conditions?

As the substrates of PALs should be in *E* configuration, we have synthesised and investigated the *E*-configured substrates. Considering the environment of the PcPAL's active pocket (PDB: 6HQF), the *Z*-configured substrates cannot be accepted by PALs. Thus, they were not studied.

2) The yield for the standard reaction decreased after 18 h and the time curve of substrate concentration showed this might result from the reverse deamination reaction (Fig. 3c). Rationales are needed to account for the fact that these reactions did not reach equilibrium after a long period of reaction time. Is it due to the concentration change of the NH₃ source in the reaction?

This is a very useful question. As depicted in Figure 3c, our data indicates that the reaction did not achieve equilibrium within the 30 h. We believe that two primary factors contributed to this observation. Firstly, prolonged exposure of the cells to a strong alkaline environment may result in the death of *E. coli* cells, consequently leading to a reduction in enzyme activity. Secondly, as mentioned by this reviewer regarding the ammonia concentration issues, inevitable ammonia leakage during the sampling process can disrupt the equilibrium of the reaction.

3) As mentioned in question 2, the reaction time will influence the final yield. For some tested substrates that showed lower yields (2a, 6a, 10a in Fig. 5a), what results will be attained after either prolonging or shortening the reaction time for these specific substrates?

Exploring different reaction conditions for various substrates indeed has the potential to alter yields. However, the low yields observed for 2a, 6a, and 10a may largely stem from intrinsic properties of the substrates themselves. Particularly for 6a and 10a, substrates bearing electron-donating substituents on the benzene ring tend to exhibit relatively lower conversion efficiencies even in the absence of β -methyl substitution (see: ACS Catalysis, 2018, 8(4): 3129-3132). Our work primarily aims to provide a universally applicable amination strategy. Therefore, throughout the manuscript, we have employed uniform reaction conditions. However, readers have the option to further optimize reaction conditions if they are interested in specific products.

4) For supplementary materials, the ^{13}C NMR spectra shown in Fig. S6/S11/S22/S26 are unsatisfactory. Please pay attention to the ^{13}C - ^{19}F coupling in fluorine-containing compounds such as substrates 2/8, products 2a/8a. Proper description of chemical shifts and coupling constants is needed in the data characterization section.

Thank you for your insightful comments and suggestions. In response to the concerns about the spectra quality, we have reanalysed the NMR data to enhance their clarity. In the revised manuscript, we have amplified the critical and interesting regions of the NMR spectra and assigned all signal peaks to their corresponding structures for each compound.

Regarding the ^{13}C - ^{19}F coupling in fluorine-containing compounds, we observed peak splitting in compounds 8/8a and 2/2a. Besides, for the ^{13}C NMR spectra in Figure S26, we did not detect the signal for the carboxyl carbon. This absence might be related to the relaxation time of the carbon nucleus.

Reviewer #2:

General comments:

The manuscript describes the extension of the substrate scope of phenylalanine ammonia lyases (PAL) for the asymmetric synthesis of β -branched aromatic α -amino acids. By the aid of computational analyses, the possible reasons behind the PAL's inability to accept β -methyl cinnamic acid (β -MeCA) as substrate have been delivered, while using a double (PcPAL-L256V-I460V) and a triple mutant (PcPAL-F137V-L256V-I460V) the successful synthesis of several β -branched phenylalanines have been performed.

The result is of high significance to the field of biocatalysis, providing the first successful application of PALs in the synthesis of phenylalanines with substituent in the β -position and the proposed molecular mechanism for the inability of the wild-type PAL to transform these β -branched substrates. Accordingly, while the manuscript employs mutant variants of the hydrophobic binding pocket of PcPAL, recently explored and mapped by mutational and structural analysis, the application of PcPAL mutants to transform the β -branched substrates is highly original in respect with the state of art. The interaction of the catalytically essential Tyr110 with the carboxyl group of the substrate, instead of assisting at the β -deprotonation, is also highly original. However in this later result, some scepticism should be formulated: considering the i) high flexibility of the Tyr110-loop, supported by its different positioning/or absence among the reported crystal structures of PcPAL (1W27, 6RGS, 6F6T, 6HQF) and the ii) limited level of accuracy of predicting the positioning of flexible loops by computational approaches, the molecular level rationale based solely on computationally assessed atomic distances is considered a weak point of the manuscript. The experimental structure (e.g. of the inactive wt-PcPAL with the β -Me-Phe) confirming the estimated distances would greatly improve the manuscript's impact and bring the ultimate support for the described molecular level mechanism.

We are very grateful also to reviewer#2 for this very positive evaluation of our manuscript emphasizing its high originality.

Due to the high flexibility of the loop region where Tyr110 is located, it is impossible to obtain a reliable crystal structure covering this loop region. Furthermore, crystal structures typically represent only one or a few states within the entire catalytic process and are insufficient to reveal the complex catalytic mechanisms of enzymes like PALs. For the investigation of loop regions, MD simulation has been proven to be a valuable tool in numerous studies. Considering these factors, we used AlphaFold2 to predict the tetrameric structure of PcPAL and its mutants for further MD simulations.

Additionally, based on the feedback from other reviewers, we re-evaluated the entire computational section and proposed that the inability of PcPAL-WT towards β -MeCA is due to its overly stable ground state in the active pocket. In the revised version, we supplemented the cMD data for the protein-ligand complex (Extended Data 2), extended the umbrella sampling window duration from 500 ps to 2 ns, and statistically analyzed the distances and angles between the amino N of MIO and the α C of the substrate during the umbrella sampling process. Moreover, we quantified the dynamic pocket volume differences among the (CA)/(PcPAL-WT), (β -MeCA)/(PcPAL-WT), and (β -MeCA)/(PcPAL-L256V-I460V) complexes and increased the number of frames for QM/MM optimization from 1 to 5. Ultimately, by comparing the substrate binding in different complexes and analyzing the interactions between Tyr110 and the substrate, we found that when β -MeCA binds to PcPAL-WT, the compact pocket squeezes β -MeCA, causing it to deviate from the normal catalytic conformation. This leads to the formation of a hydrogen bond between Tyr110 and the carboxyl group of β -MeCA (Figure 2f), which further stabilizes this unreasonable conformation. This supports our hypothesis that the inability of the reaction to occur is due to the stability of the ground state.

Overall, systematically and accurately elucidating the molecular mechanism by which β -MeCA cannot be formed by PcPAL-WT is very challenging. In our study, through systematic computational analyses, we suggest that substrate ground state stability is a very likely and important factor determining whether the reaction can occur or not.

While the methodology is sound, in several cases the results interpretation, methodology development and description must be improved that the work to be reproduced.

In this context major clarifications/improvements should be provided for a novel round of evaluation.

Major issues:

1. Introduction part, paragraph starting with : "In previous reports, PALs from.... " - lit 36-40 are not about engineering these PALs, as stated by the phrase, only two of cited references deal with PAL engineering. Lit 33-36 is not exhaustive, is only selection from the literature dealing with these PALs, and they apparently do not share a common selection criteria (e.g. papers dealing with their substrate scope description or papers reporting their first discovery). At this part, references and their selection should be improved.

Regarding the comment that "Literature 33-36 is not exhaustive," we have replaced them with more representative references in the revised manuscript that are related to their first discovery and substrate scope descriptions. Additionally, for the comment that "Literature 36-40 is not about engineering these PALs," we carefully selected and added six new references specifically related to the engineering of hydrophobic binding pockets (numbered with 40 to 45 in the revised manuscript).

2. Related to Figure 2c and 2d:

- Fig 2d should present the substrate in the catalytic site at the same size as Figure 2c, the cartoon background makes it difficult to follow the details.

We have changed the original Fig 2d into a clearer version according to this suggestion. In accordance to the comments from the other reviewers, we have put these IGM results in the Extended Data 3.

- please label important residues which are mentioned in the Results section, part including the discussions related to Fig. 2c and Fig. 2d.

The same is valid for Fig. 6a

We have now labelled the important residues, including M10, R354, N384, N260, L137 and F138. The new version of these IGM results can be found in Extended Data 3.

Residue R354 should also appear as reference for its correct positioning for the fixation of the substrate's carboxyl group (including in Fig. 3 and Extended data 1.) The distances related to this residues should also be monitored, presented within the selected docking results.

We updated Figure 3a and Extended data 1, in which R354 and the corresponding distances were marked. Besides, we also calculated the distances between guanidine C of R354 and carboxyl C1 of substrates (Extended Data 2. c-d) during cMD simulations.

3. From Figure 3a, L206 does not seem to locate near the β -methyl substituent of the substrate, also the data from literature (lit 44 from the references) argue that this residue is in close proximity to the ortho-position of the substrate's phenyl ring.

We apologize for the mistake and have now revised the sentence to: "L256 and N260 are located near the β -Me group, while the others are around the phenyl ring of β -MeCA."

In Figures with the catalytic site the R354 should also appear as reference for its correct positioning for the fixation of the substrate's carboxyl group.

We have made the changes as described for question 2.

4. Section Rational engineering of PcPAL for the amination of β -MeCA, phrase: Additionally, we also investigated three other mutants, previously established in our laboratory (E484D, F137V-I460V40, and F137A-I460A40) to explore their potential.

- please mention the origin and the previous utility of double mutant F13V-I460V/A

We have included the following sentence in the revised manuscript: "Notably, the latter two mutants, developed by Bencze's group (numbered with ref. 41: Filip, A., et al. *ChemCatChem*, 2018, 10, 2627-2633), were investigated for their ability to catalyze the amination of bulky substrates, such as styrylacrylate, for the synthesis of bulky arylalanine."

5. Figure 4 - Provide in ESI the original chromatograms.

Legend of Figure 4a - What is the difference between v and vi? - in each case deamination of (2R,3R)- β -MePhe is written in the figure caption.

We have re-conducted our experiments and replaced the original Figure 4 with the original chromatograms in the revised version of the manuscript. In the original submission, the difference between v and vi was that v represents the (2R,3R)- β -MePhe standard, while vi represents the result after the deamination reaction using (2R,3R)- β -MePhe as the substrate. This distinction has been clarified in the revised manuscript.

6. Section "Development of a whole-cell biotransformation process":

a) Why 5M ammonia and 4 M ammonium carbamate buffer was tested?

In the PcPAL-mediated reactions other optimal ammonia concentrations are reported, such as 2 M ammonium carbamate, 4 M NH₄OH. Further ammonium carbamate might release 2 eq. of ammonia, thus it would be equivalent to 8 M NH₄OH, in this way it is hard to compare it with the 5M NH₄OH buffer. Please provide some rationale for the employed ammonia concentrations.

According to the previous reports, the choice between NH₄OH buffer and ammonium carbamate buffer can arise differences in the amination reaction of PALs. Our initial aim was to determine which buffer was more suitable for PcPAL. Therefore, we selected these two commonly used buffers as reported in the literature (see: *Angew. Chem. Int. Ed.* 2014, 53, 4652-4656; *Catal. Sci. Technol.* 2016(6), 4086-4089; *Protein Engineering, Design & Selection*, 2010, 23(12): 929-933).

Our experimental results demonstrated that PcPAL performed better with 5 M NH₄OH buffer compared to 4 M ammonium carbamate buffer (Figure S4). Just as the reviewer's comments here, 4

M ammonium carbamate should theoretically be equivalent to 8 M NH₄OH and thus expected to perform better than 5 M NH₄OH. However, our results indicated that NH₄OH was more suitable for PcPAL. Consequently, we chose to use 5 M NH₄OH in the subsequent experiments and did not pursue further detailed comparative experiments.

b) “ .. we found that 5 M ammonium hydroxide buffer (NH₄OH buffer, pH 10) performed better than 4 M ammonium carbamate buffer (NH₂CO₃NH₄ buffer, pH 10) (Fig. S4).” and later: “...gave significantly higher conversions than using clarified lysate, presumably due to higher enzyme stability in vivo under the same catalytic conditions (Fig. S4).” - please include somewhere the conversion values related to chromatograms from Fig.S4.

In our work, we aimed to explore the impact of different catalytic forms on the reaction. In Figure S4, the efficiency of whole-cell conversion is noticeably higher than that of the lysate catalytic form. Given this pronounced disparity, which intuitively indicates the superiority of whole-cell conversion efficiency over the lysate form, we didn't further quantify the conversion efficiency of the lysate, and this will not affect our conclusions.

7. Section “Elucidating the molecular basis of PcPAL-L256V-I460V for catalyzing β-MeCA”

It is stated that the protein structure of the mutant variant was predicted by AlphaFold. The build-up of the MIO-group how was achieved by using AlphaFold? And Tyr110-loop? Please comment and provide details in the ESI related to this model built-up, since according to the reviewers experience, the MIO-group cannot be built up by Alpha-Fold and also the Tyr-loop is modelled too tightly or loosely within the catalytic site.

We made this clear in the 'Method' section of the revised manuscript. Overall, the protein structure was predicted using AlphaFold 2, while the MIO structure was derived from a resolved crystal structure. We extracted the MIO-group from the crystal structure (PDBID: 6RGS) and re-edited it in PyMOL. Subsequently, the MIO structure was optimized using the software ORCA 5.0.4 at the r2SCAN-3c theoretical level to ensure the accuracy of its geometry and electronic structure. Next, RESP2 charge fitting was performed using Multiwfn_3.8_dev, and the MIO group was further parameterized using the Antechamber tool based on Amber atom type. Finally, missing parameters were completed using the parmchk2 tool, and the results were visually inspected and subjected to dynamic testing.

8. Kinetic data for the transformation of β-MeCA or the reverse reaction catalyzed by the PcPAL-L256V-I460V should be also assessed, in order to position within the state of art the catalytic efficiency of the mutant variants towards these substrates.

Due to the presence of high concentrations of ammonia in the amination reaction system, determining of the kinetic parameters of the amination reaction is often not very precise. Consequently, many significant studies focusing on amination reactions have not reported the kinetic constants of enzymes during the amination process (see: Angew. Chem. Int. Ed. 2024, e202406008; ChemCatChem, 2018, 10(12): 2627-2633; Angew. Chem. Int. Ed. 2015, 54, 4608–4611; ACS Catalysis, 2018, 8(4): 3129-3132). Similarly, in our case, our primary goal was to develop an engineered amination platform, and we initially did not investigate the kinetic parameters. However, as suggested by this reviewer, to evaluate the catalytic efficiency of the mutant, we measured the kinetic constants of the deamination reaction using the mutant PcPAL-L256V-I460V with (2S,3R)-β-MePhe (Numbered with 1a).

Enzyme assays were conducted in triplicate at 30°C using 96-well UV-microplates. The reactions were carried out in 50 mM PBS buffer (pH 8.0) and the mutant enzyme (PcPAL-L256V-I460V) was maintained at a constant concentration of 0.6 μM, with eight different substrate concentrations ranging from 0.015625 to 1.5 mM. Kinetic measurements were performed using UV spectroscopy to monitor the production of β-MeCA over a 10-minute period at a wavelength of 260 nm. Kinetic constants were determined from Michaelis-Menten plots via nonlinear regression analysis.

The results were as follows: $K_m = 0.7304$ mM, $K_{cat} = 3.811$ min⁻¹, and $V_{max} = 2.896$ μmol/min. The relatively high K_m value indicates the low affinity of (2S,3R)-β-MePhe to the mutant, suggesting that the product can more easily leave the catalytic pocket during the amination reaction, thereby facilitating the amination reaction.

Figure R1. Michaelis-Menten curves for the ammonia elimination from (2S,3R)-β-MePhe catalyzed by PcPAL-L256V-I460V; measured in triplicate.

Data, methods within supporting information should be seriously improved in terms of quality:

9. Supporting information Section 1.1. - Reference 2 is too general to be acceptable, please replace with more specific references or provide page-numbers from this book.

Thank you for your valuable comments. We acknowledge that Reference 2 was incorrectly cited, as it pertains to *Streptomyces* operations. We apologize for this oversight. In the revised version of our manuscript, we have removed Reference 2.

10. Supporting information, section Optimization of the in vivo amination reactions

“...These reactions were placed at 250 rpm and 30 °C for 24 h. Then 100 μL samples were extracted and supplemented with 100 μL aq. H₂SO₄ (10% v/v) to stop the reaction. Subsequent centrifugation (12,000 × g, 4 °C, 2 min) separated the supernatant, which was then subjected to filtration and directly injected into an HPLC apparatus for analysis.”

Stopping the reaction with acidification, followed by filtration might lead to serious errors in the conversion values. The cinnamic acid substrates under acidic conditions have low water solubility, might form micro suspensions, which together with cell debris are partially removed by filtration. This issue was already observed and reported during the optimization of the work-up procedure of the whole-cell PAL biotransformation within reference 46. This might be and was alleviated by stopping the reaction by adding 50% MeOH, followed by filtration (with or without adding acid – in function of the employed HPLC method) and injection to HPLC. Please check if by this later sample preparation method the same conversions are obtained than the actual conversions.

In our study, the conversion was calculated based on the formation of products. During our experimental procedure, we observed that many substrates (such as S12) have very low solubility in methanol, and the products were almost not soluble in methanol solution but exhibit good solubility in acidic solution. Besides, similar stop methods were used in some other papers (see: *Angew. Chem. Int. Ed.* 2015, 54, 4608–4611; *Angew. Chem. Int. Ed.*, 2021, 60(32): 17680-17685). Therefore, to accurately detect the yield of target products, we chose to stop the reaction by adding HCl to the reaction system to mainly focus on the amination products of this study. To enhance clarity, we have revised the y-axis title of Figure 3b and 3c from ‘conversions’ to “product concentration”.

11. Supporting information page 10 – please provide the original and full chromatograms, the region with the peaks might be zoomed as it appears currently, however the original chromatograms are mandatory.

Thank you for your comments. We have now added all the corresponding original chromatograms into the SI.

12. Supporting information – in all cases >99.5% ee values are reported, however in several cases, e.g. 5a, 6a, 8a, the intensity of the peaks relatively to the baseline noise is excluding that such high ee values could be determined. The noise derived from the baseline controls the detection limit of the undesired enantiomer/diastereomer, which should be taken into account. In this sense for the accurate ee measurements probably peaks of higher area would be requested in fact in all cases.

To ensure the accuracy of our results, we repeated the derivatization experiments using purified products. After re-analyzing the samples with HPLC, we excluded noise signals derived from the baseline. Just as mentioned for question 11, the original chromatograms are now provided in the SI. The repeated results confirmed that the conclusions presented in the original manuscript are reliable.

13. NMR spectras - Figures should be amplified on the interesting region, and also during data interpretation and in Figures signals must be linked to the molecules protons.

Some ¹³C-NMR measurements need more scans, the signal intensities are very low, eg. – Figure S6, Figure S11.

In figure S15, ¹H-NMR spectra seems to contain besides the specific signals of the molecule several additional signals, please explain or improve compound purity.

In the new manuscript, we have amplified the important and interesting regions of the NMR spectra and assigned all signal peaks to their corresponding structures for each compound. For the concerns about the spectra quality, we have reanalysed the data using a 600 MHz NMR device to enhance their clarity. The new NMR results showed much better quality than the previous ones. Additionally, after enhancing the compound purity, the additional signals in Figure S15 have been eliminated.

14. Figure S1 - standard deviations are needed.

The use of calibration curve based solely on the product's signal, might provide large standard deviations in the conversion values, due the effect of small daily differences in analyte and analysis conditions on the peak area of individual signals. This is usually alleviated by the use calibration curves based on relative intensities of both relative substrate and product or the use of internal standard. In the absence of these, please provide information what are the standard deviations of conversion assessments when the same sample is injected in duplicate or triplicate. This might be surprisingly large, also considering the issues mentioned in point 2.

When establishing the standard curve, we conducted three replicate measurements for each sample, and the standard deviations are now provided in the revised version of Figure S1. The minimal differences observed among the replicates indicate the reliability and stability of our system.

15. Figure S3 – several chromatograms show additional peaks of non-negligible intensities (e.g. b, e, f) of which appearance and nature should be commented or explained.

We have addressed your concern regarding Figure S3 by adding more data in Figure S3. However, as there are no figures numbered with 'f' in Figure S3, we believe you may be referring to Figure S2.

For Figure S2, we repeated all the amination reactions again and analysed on a different HPLC. we observed that the additional peaks detected could also be identified in blank samples containing only the substrates. Besides, we also list the UV-absorption spectra of the substrates and products in figure S2 B. As the additional peaks' UV- absorption were different from our products, we thought they may have been caused by impurity or decomposition of the substrates.

16. "Figure S32 . Overlap check of specific windows with histograms" – please describe what represents this image, X-axes what represent, etc.

This figure presents a comparison of overlap checks and histograms for specific windows. The X-axis represents the reaction coordinate, while the Y-axis shows the corresponding frequency. Through this figure, we can visually observe the overlap of different windows on the reaction coordinate and the distribution of data within each window. This is crucial for verifying the continuity of the reaction pathway and the rationality of window settings, thereby ensuring the accuracy and reliability of the calculations.

17. The authors should specify the program used for molecular docking with details, such as the grid size, etc. The software mentioned in the SI, is just a GUI. (correct name is autodock tools 1.5.6)

We have added more details in the 'Method' section of the revised manuscript as follows: The docking experiments were carried out using AutoDock Tools (version 1.5.6), with the grid size restricted to a 10x10x10 Å box centered on the midpoint between the MIO residue and Tyr110 of the tyrosine loop.

Minor issues:

1. Introduction part, paragraph starting with : "In previous reports, PALs from.... " insert space between word '...variabilis' and 'their'

The correct name of the strain from which AvPAL is derived is *Anabaena variabilis*. Therefore, we have removed the erroneous word "their" in the revised manuscript.

2. pg. 9, Phrase: "Through analyzing the crystal structure of PcPAL (PDB:6HQF), we found that L256 is positioned in close proximity to the β-methyl group of the substrate (Fig. 3a)" - I guess this is based on the analysis of a docking result, with the β-methyl-Phe docked into 6HQF, not of the crystal structure as it might be interpreted from the text.

It is correct that our analysis was based on docking results. We have revised the text from "Through analyzing the crystal structure of PcPAL (PDB: 6HQF)" to "Through analyzing the conformation of β-MeCA within the docking result."

3. Please correct PcPAL to its correct form *PcPAL*

All the PcPAL were changed into *PcPAL* in the revised version. Besides, we also changed RgPAL, RtPAL and AvPAL into *RgPAL*, *RtPAL* and *AvPAL*.

4. Figure 3a – please mention error bars are derived from triplicate or duplicate experiments.

We have added the statement "Error bars are derived from triplicate experiments and were created using GraphPad Prism 8" at the end of the Figure 3 legend.

5. First sentence of Section "Development of a whole-cell biotransformation process", instead of literature 48, 49, more representative would be the recently optimized whole-cell procedures using PcPAL and/or its mutant variants, references 46 and 45 from the manuscript.

For this part, we have updated the manuscript to cite the original references numbered as 45 and 46.

6. Supporting information, pg. 11-16– correct the NH₂- group in the structure of the compounds (number in subscript).

We have revised the structures accordingly by changing the NH₂ into NH₂ in the structures. Besides, we also changed the NH₂ in Figure 1c into NH₂.

7. Figure S31 – highlighted residues and the MIO-group should be labelled

We have labelled the key residues (including Y110, L256 and I460) and the MIO-group in the new version SI and changed the number of the original Figure S31 into Figure S32.

Reviewer #3:

In this work, Sun et. al targeted the challenge of the asymmetric synthesis of β -branched aromatic α -amino acids using engineered phenylalanine ammonia lyases (PALs). This biocatalytic outcome is highly significant because it achieves the desired synthesis through direct amination with a high diastereoselectivity, enantioselectivity, and yield without requiring additional cofactor regeneration system. This work also demonstrates a high level of innovation as it integrates computational tools to guide experimental mutation discovery and to elucidate the mechanisms behind catalysis. Below, we listed critical comments regarding the organization, necessity, and scholarly presentation of the computational components. We hope these comments can help improve the quality of the draft to a publication level:

We are very grateful also to reviewer#3 for this very positive evaluation of our manuscript and also for highlighting “the high level of innovation” of our work. We are especially grateful that publication of our work in *Nature Communications* has been suggested after revision.

Major issues:

1. As a specialist in computational modeling, I find the “In silico analyses of PcPAL towards β -MeCA” to be not convincing. In this section, the central point of computational investigation is the origin behind which PcPAL can catalyze CA but not β -MeCA, which is indeed an important question. Based on existing theories, reasonable hypotheses behind the lack of activity could be differences in TS stabilization (Houk, Warshal), ground state destabilization (Hershlag), substrate positioning dynamics (Benkovic), binding, and so on. However, it was not clear to me what fundamental hypothesis the authors made when conducting the computational analysis, albeit that substantial efforts have been spent on comparing the geometric/conformational differences between two complexes using docking, classical MD and QM/MM. I suggest the authors to articulate their hypothesis and frame their computational results around it. For example, the substrate positioning dynamics may be hypothesized to control the selectivity. As such, all comparisons on the reaction coordinate distribution will make sense, as long as the authors provide or cite evidence to justify that TS barriers are similar. Instead, the authors may hypothesize TS barrier difference is the key cause, but how to relate the ground state/reactive conformation to TS is another point of justification.

As suggested, we propose a reasonable hypothesis. To achieve this, we have restructured the entire computational section, supplementing it with cMD data of the protein-ligand complex, and extended the umbrella sampling window from 500 ps to 2 ns. We also performed statistical analysis on the distance and angle between the amino N of MIO and the α C of the substrate during the umbrella sampling process and analyzed the pocket volume during this process. Finally, we also increased the QM/MM optimization from 1 frame to 5 frames. After reorganizing and supplementing our computational process, we propose that the primary reason for PcPAL's inactivity toward β -MeCA is due to the ground state.

From our cMD simulation data, through examining the hydrogen bond interactions between the substrates and amino acids within the active pocket (Supplementary Data 5), it was found that in the (β -MeCA)/(PcPAL-WT) complex, the carboxyl group of β -MeCA forms a high-frequency hydrogen bond with Tyr110, appearing in 99% of the recorded 5000 snapshots (Supplementary Data 5).

Additionally, we quantified the dynamic pocket volume differences of (CA)/(PcPAL-WT) and (β -MeCA)/(PcPAL-WT) complexes, finding that the pocket cavity of (CA)/(PcPAL-WT) was significantly larger than that of (β -MeCA)/(PcPAL-WT) complexes. By further comparing the substrate binding modes by QM/MM, we reasonably inferred that when PcPAL-WT binds β -MeCA, the tight pocket cannot adequately accommodate the methyl substrate, causing it to deviate from the catalytic conformation and leading to the formation of a hydrogen bond between Tyr110 and the carboxyl group of β -MeCA (see figure 2f), which further stabilizes the binding of WT and β -MeCA.

Overall, both experimental and computational results demonstrate the reliability of our findings, indicating that the stable ground state of β -MeCA within PcPAL-WT might be the reason for its inactivity.

2. At the end of the computational analyses, the authors propose the molecular principle of mutagenesis. Though appearing reasonable, many statements involved in the chain of reasoning appears inconclusive. First, the docking studies are good for providing candidates for MD and QM, but are not quantitative enough (more than “proved insufficient”) to judge selectivity and conformational differences. Second, the umbrella sampling involves limited conformational sampling and can not “strongly indicate” the lack of PcPAL activity toward β -MeCA. I am sure that authors realize this and made more substantial sampling in the subsequent section, but it does not help remedy the logical issue in umbrella sampling. Third, IGM analysis is typically used to qualitatively show what interactions are important. The conclusion from the IGM analysis is pretty vague. Fourth, the authors mentioned about different hydrogen bonding patterns between two complexes derived from QMMM calculations, which is pretty interesting, but they fail to justify the number of snapshots involved in the calculations, and which one involves lower energies or major conformational cluster, etc. Overall, I would suggest rewrite the computational section to enhance the internal logic.

We have rewritten this section accordingly to the suggestions as follows:

1) For ‘First, the docking studies are good for providing candidates for MD and QM, but are not quantitative enough (more than “proved insufficient”) to judge selectivity and conformational differences.’

We totally agree. In the revised version, we have changed the corresponding sentences into ‘As docking results were not quantitative enough to judge selectivity and conformational differences, we further performed conventional molecular dynamics (cMD) simulations.’

2) For ‘Second, the umbrella sampling involves limited conformational sampling and can not “strongly indicate” the lack of PcPAL activity toward β -MeCA. I am sure that authors realize this and made more substantial sampling in the subsequent section, but it does not help remedy the logical issue in umbrella sampling.’

We agree that umbrella sampling involving a limited number of conformations cannot definitively indicate PcPAL's lack of activity towards β -MeCA. To clarify this, we have revised the sentence to: "The results of umbrella sampling indicated that the likely reason for PcPAL's inactivity towards β -MeCA is the excessive stabilization of β -MeCA's ground state due to the crowded pocket" at the end of second paragraph of ‘*In silico* analyses of PcPAL towards β -MeCA’ part. Additionally, we have extended the window duration of umbrella sampling from 550 ps to 2 ns. While this does not rectify the inherent logical flaws of umbrella sampling, it will enhance the accuracy of our PMF curve.

3) For ‘Third, IGM analysis is typically used to qualitatively show what interactions are important. The conclusion from the IGM analysis is pretty vague.’

Through our IGM analysis of different complexes, we can draw the following conclusions: β -MeCA is relatively crowded in the active site pocket of PcPAL-WT and the strong interaction between Tyr110 and carboxyl group of β -MeCA is important. In contrast, CA in the active site pocket of PcPAL-WT and β -MeCA in the mutant pocket are relatively flexible and do not form a hydrogen bond with Tyr110.

4) For ‘Fourth, the authors mentioned about different hydrogen bonding patterns between two complexes derived from QMMM calculations, which is pretty interesting, but they fail to justify the number of snapshots involved in the calculations, and which one involves lower energies or major conformational cluster’.

In our previous simulations, we had chosen the first cluster generated by clustering, out of a total of three clusters. However, in our latest computations, we clustered a total of five clusters and optimized all of them. Detailed methods are provided in the ‘Methods’ section. Additionally, we have uploaded the optimized files as zip files which include the optimized XYZ files (Supplementary Data 7).

The reason for uploading XYZ files is that they offered higher coordinate precision than PDB files and are the default format for recording coordinates in the ASH program. Consistent with our previous results, stable strong hydrogen bonds between TYR110 and the carboxyl group of β MeCA could be observed in all clusters in (β MeCA)/(PcPAL-WT) complex.

3. The authors proposed two pieces of engineering principles from their computational studies – they are “alleviating the stringent active pocket of PcPAL to enhance the conformational flexibility of β -MeCA analogs” and “inducing reactive conformations by engineering not only the residues around the β -methyl group of β -MeCA, but also those around its phenyl ring”. Changing active site pocket is a common practice in rational protein engineering, and many of these insights can be inferred merely from cavity analysis or MD trajectory analysis without extensive umbrella sampling and QMMM calculations. As one example, the key conclusion about the H-bond can be obtained by just analyzing the trajectory of the cMD (the distance distribution as a histogram). The authors would want to justify the necessity of using such an extensive set of computational protocols. In particular, why the same type of insight is difficult to gain without QMMM or umbrella sampling.

cMD can indeed provide similar conclusions for rational protein engineering. However, our study aims not only to gain insights into engineering strategies but also to elucidate the underlying reasons behind this longstanding problem. Our initial cMD attempts revealed that β -MeCA and CA have similar binding modes. The RMSD of the backbone atoms and the RMSD of the heavy atoms within 7 Å of the substrates during cMD simulations were similar in both (CA)/(PcPAL-WT) and (β -MeCA)/(PcPAL-WT) complex systems. Additionally, the distance between the MIO's N atom and the substrate's α C was similar, thus we could not deduce the reason for the inactivity of PcPAL-WT towards β -MeCA directly from the cMD simulation results (Extended Data 2).

Moreover, by selecting the distance between MIO's N atom and α C as the reaction coordinate and fitting the free energy curve using WHAM, we could observe a relationship between the distance of MIO's atom to α C and energy changes. However, these details are not easily obtained by cMD simulations. Using umbrella sampling and WHAM-fitted PMF, we could observe energy variations and infer differences in binding dynamics between the wild type and mutants, and therefore this helped to understand the molecular mechanism underlying the PcPAL-WT's inactivity towards β -MeCA.

Furthermore, we followed the suggestion and used fpocket to analyze the volume changes of the pocket cavity throughout the trajectories. We calculated the pocket volume variations during umbrella sampling by extracting 20 frames from each window (a total of 1800 frames) for pocket cavity volume calculation. Since the coordinates of the umbrella sampling windows are not strictly continuous, we produced scatter plots and distribution diagrams (Fig 2c, 2d, and 6b). The results showed that the (CA)/(PcPAL-WT) and (β -MeCA)/(PcPAL-L256V-I460V) complex systems exhibited larger pocket volume, while the pocket of (β -MeCA)/(PcPAL-WT) appeared more compact.

Minor issues:

1. (line 130) Please describe how are the starting structures for US selected from the cMD. Please include this structure in the SI. (PDB files in .zip and figures for the active conformations)

We have made this clear in the ‘Method’ part: ‘The initial coordinates for umbrella sampling were derived from equilibrated conformations obtained during the MD phase, followed by three independent 10 ns NPT simulations to serve as the starting points for the three parallel sets of umbrella sampling.’ We also uploaded the initial structures for the umbrella sampling as PDF files in the zip files (Supplementary Data 6).

2. For texts between line 124 and 145 describe the process of discovery but do not support any statements. They are good fit for SI.

We have re-written this paragraph according to this suggestion. The results of IGM were moved to Extended data 3.

3. (line 150) The setup of this 2nd round of US simulation is vague. What is the starting structure? How is the pre-reactive state defined and modeled?

We have re-written this section in the 'Method' part : 'In the pre-reaction state sampling, we first read the coordinates corresponding to the lowest energy point from the PMF curve obtained through umbrella sampling. Then we performed a 2 ns classical molecular dynamics (cMD) simulation starting from these coordinates, recording snapshots every 1 ps. Finally, we used cpptraj to perform clustering analysis on these snapshots, setting a total of 5 conformational clusters.'

4. I suggest moving most of the IGM analysis to SI and leaving only the insights about the interactions between Tyr110 and beta-MeCA.

We have made changes according to this suggestion.

5. The details of IGM setup appears missing. The author needs to describe the detailed selection of fragments in the Method section.

We have rewritten the section related to IGM analysis in the Method section: 'The Independent Gradient Model (IGM) of Multiwfn_3.8_dev was utilized to investigate attractive interaction between clustered conformations of the substrate and protein with isosurface value of 0.1. The IGM interaction regions and color-mapped isosurface diagrams were obtained using the VMD 1.9.3 program.'

6. Line 167, the QM/MM optimization uses only one frame from the MD or US. The authors may consider moving the QM/MM part to SI and just analyzing the trajectories from cMD for distance distribution, unless they would want to afford QMMM optimization on more snapshots.

have increased the number of optimized frames and the distribution of these optimizations can be found in Table S6.

7. The 2nd part of the computational study from line 363 suffers from the same problem I commented on for the 1st part in comment #10. Please address this accordingly. The mutants help the catalysis by ground state destabilization (i.e.: the reactant state won't form the H-bond).

We have adopted the ground state destabilization hypothesis and made revisions in the 'In silico analyses of PcPAL towards β -MeCA' sections in the revised manuscript.

8. It will be helpful to readers if the authors also compare the performance of WT in Figure 5.

As the WT cannot aminate β -MeCA, we did not test its activity towards other β -MeCA analogs.

Reviewer #4:

Thanks a lot for your comments and suggestions which have been very helpful to revise our manuscript.

REVIEWER COMMENTS

Reviewer #1 (Remarks to the Author):

In this revised manuscript, Bornscheuer and coworkers addressed all my questions. I strongly support the publication of this work in Nature Communications. Before the publication of this paper, the following recent publications that are relevant to the synthesis of beta-methylphenylalanine derivatives should be cited:

1. Stereoselective amino acid synthesis by photobiocatalytic oxidative coupling, Nature 2024, 629, 98–104. (photobiocatalytic C-C bond formation to prepare beta-methylphenylalanines)

2. Transaminase-Catalyzed Synthesis of β -Branched Noncanonical Amino Acids Driven by a Lysine Amine Donor. J. Am. Chem. Soc. 2024, 146, 23, 16306–16313 (dynamic kinetic asymmetric transformation to prepare beta-methylphenylalanines, related to ref. 24 of the authors' manuscript)

With these updates, the authors are advised to update Figure 1 accordingly to include these results.

Overall, this is an excellent manuscript solving a long-standing problem in biocatalysis based on the MIO cofactor. I congratulate the authors on their achievement detailed in this paper.

Reviewer #2 (Remarks to the Author):

The manuscript have been substantially revised and improved during the first revision round, the authors fulfilled and addressed the majority of the requests.

Still one major issue remained from my side:

1) HPLC chromatograms for 1a-10a from ESI: as requested before, please provide the original and full chromatograms, usually directly taken out from instrument in form of reports. The current forms does not represent original research data, which are highly recommended to be used as supplementary data. Please comment why the retention times

for the derivatized 1a-10a significantly changed within the revised version, despite no modification in the used HPLC method (described in pg S10).

Please also consider the following minor issues:

1) Pg. 10. Newly introduced section:

“Notably, the latter two mutants... to catalyze the amination of bulky substrates, such as styrylacrylate, for the synthesis of bulky arylalanine”

Incorrect phrase ending, I recommend the following form, also reflecting more the cited work: “.....to catalyze the amination of bulky substrates, such as styrylacrylate and biarylalanines.”

2) Legend Figure 3. “Conversions were calculated ...” please correct to “ Product concentrations were calculated...”

3) I would argue against the response related to the issues of the used ammonia-buffer:

“According to the previous reports, the choice between NH_4OH buffer and ammonium carbamate buffer can arise differences in the amination reaction of PALs. Our initial aim was to determine which buffer was more suitable for PcPAL. Therefore, we selected these two commonly used buffers as reported in the literature (see: *Angew. Chem. Int. Ed.* 2014, 53, 4652–4656; *Catal. Sci. Technol.* 2016(6), 4086-4089; *Protein Engineering, Design & Selection*, 2010, 23(12): 929-933).”

Besides and after the appearance of the cited manuscripts several studies optimized the reaction medium, ammonia source and ammonia concentration of the PAL mediated reactions (including lit 41,42,53 cited in the manuscript and others), which highlighted that optimal ammonia concentration might vary from substrate to substrate and ranges between 2-4 M carbamate and/or 4-8 M ammonia.

But, nonetheless, to be able to compare the results from the reactions employing different ammonia sources one should try to use at least the same ammonia concentration, which didn't happen in this work. In case if the reaction cannot be repeated in 8 M ammonia or 2 M ammonium carbamate, then please leave out their comparison and mention that 5 M ammonia was used as in other studies.

4) short discussion, sentence of the kinetic data should be introduced within the main manuscript, (e.g. above part including Figure)

Reviewer #3 (Remarks to the Author):

The authors have made changes on the manuscript, but some prior issues remain, particularly regarding the organization, the logic flow, and conclusiveness of the computational modeling. Despite a strong experimental outcome, this research work still appears to be weakened by the lack of hypotheses and justification in data interpretations, leading to apparently unsupported conclusions. To enhance the paper, the author is advised to reconsider what computational studies really matter and make sure the discussions are grounded from molecular theory or physical chemistry basis, rather than discussing most of the data based on their chemical intuition. As such, we provide more specific comments here, with the hope to help the authors further improve the quality, clarity, and impact of the manuscript.

1. The author needs to add a scheme and an introduction about the proposed mechanisms of PcPAL. This will give readers who do not know PAL chemistry a better context of its subsequent discussions.
2. The docking results cannot support conclusions such as “analogous conformations”. The author should remove them and review them using the conformational ensemble obtained from the cMD simulation.
3. The author analyzed “the RMSD of backbone atoms”, “RMSD of heavy atoms within 7Å of substrates”, “distances to the R354 residue”, and “the amination reaction distance” based on the cMD simulation. However, the relevance of these metrics to the catalysis is not justified. Replacing the substrate in the MD simulation mainly perturbs the substrate positioning dynamics. But these changes are not directly reflected from those RMSD measurements of the greater protein regions (i.e.: the entire backbone and residues region of 7Å) The measurement of the “distances to the R354 residue” and “the amination reaction distance” comes out a bit sudden without any prior discussion of why these metrics are used. Some discussions appear ungrounded or irrelevant, for example, “β-MeCA and CA have similar binding modes” and “we could not deduce the reason for the inactivity of PcPAL-WT towards β-MeCA directly from the cMD simulation results”. I suggest removing large part of these irrelevant discussions and keep only the parts that are conclusive from both physical chemistry and biocatalysis perspective.
4. The author repetitively uses “the amination reaction distance” in cMD analysis and umbrella sampling. Although the description is intuitive, the physical meaning behind these

metrics is vague under Molecular Mechanics context. Calculating binding energy or activation barrier from the umbrella sampling or QM/MM under well-established physics models will be a more legit and natural way to go. Please provide the theoretical justification of how this distance is related to the catalysis. Alternatively, the authors may consider removing most of these discussions if the justification is weak.

5. Overall, if some computational studies such as umbrella sampling, QM/MM and metrics such as “the amination reaction distance”, “backbone RMSD” cannot be justified, their removal may not be a bad choice.

Reviewer #4 (Remarks to the Author):

I co-reviewed this manuscript with one of the reviewers who provided the listed reports.

This is part of the Nature Communications initiative to facilitate training in peer review and to provide appropriate recognition for Early Career Researchers who co-review manuscripts.

REVIEWER COMMENTS for manuscript NCOMMS-24-15772-T

Reviewer #1 (Remarks to the Author):

In this revised manuscript, Bornscheuer and coworkers addressed all my questions. I strongly support the publication of this work in Nature Communications. Before the publication of this paper, the following recent publications that are relevant to the synthesis of beta-methylphenylalanine derivatives should be cited:

1. Stereoselective amino acid synthesis by photobiocatalytic oxidative coupling, Nature 2024, 629, 98–104. (photobiocatalytic C-C bond formation to prepare beta-methylphenylalanines)
2. Transaminase-Catalyzed Synthesis of β -Branched Noncanonical Amino Acids Driven by a Lysine Amine Donor. J. Am. Chem. Soc. 2024, 146, 23, 16306–16313 (dynamic kinetic asymmetric transformation to prepare beta-methylphenylalanines, related to ref. 24 of the authors' manuscript)

With these updates, the authors are advised to update Figure 1 accordingly to include these results.

Overall, this is an excellent manuscript solving a long-standing problem in biocatalysis based on the MIO cofactor. I congratulate the authors on their achievement detailed in this paper.

Thank you for your suggestions and comments, which have significantly contributed to the improvement of this manuscript. According to your suggestions, we have added the two references to the introduction and updated Figure 1 accordingly in the revised version.

We are of course very pleased that publication in Nature Communication has been strongly supported.

Reviewer #2 (Remarks to the Author):

The manuscript have been substantially revised and improved during the first revision round, the authors fulfilled and addressed the majority of the requests.

Still one major issue remained from my side:

- 1) HPLC chromatograms for 1a-10a from ESI: as requested before, please provide the original and full chromatograms, usually directly taken out from instrument in form of reports. The current forms does not represent original research data, which are highly recommended to be used as supplementary data. Please comment why the retention times for the derivatized 1a-10a significantly changed within the revised version, despite no modification in the used HPLC method (described in pg S10).

Thanks a lot for your comments. We are happy to learn that we have fulfilled and addressed the majority of the points raised by reviewer#2 for the first revision round. We have now placed the original and full HPLC figures in the newly revised Supplementary Information. As the new data was repeated by the HPLC method, there were some differences from the previous version. We have thus adapted the description of the HPLC method, see p. S10: "The derivatization reactions were analyzed using a Welch Ultimate XB-C18 column (5 μ m, 250 \times 4.6 mm) at a flow rate of 1 mL min⁻¹ and detected at 340 nm over a 27 min gradient program with water containing 0.1% formic acid (eluent A) and MeOH (eluent B): T = 0 min, 30% B; T = 0 – 20 min, 95% B; T = 22 min, 95% B; T = 23 - 27 min, 30% B."

Please also consider the following minor issues:

1) Pg. 10. Newly introduced section:

“Notably, the latter two mutants... to catalyze the amination of bulky substrates, such as styrylacrylate, for the synthesis of bulky arylalanine”

Incorrect phrase ending, I recommend the following form, also reflecting more the cited work: “.....to catalyze the amination of bulky substrates, such as styrylacrylate and biarylalanines.”

We have made these changes accordingly.

2) Legend Figure 3. “Conversions were calculated ...” please correct to “ Product concentrations were calculated...”

We have made these changes accordingly

3) I would argue against the response related to the issues of the used ammonia-buffer:

“According to the previous reports, the choice between NH₄OH buffer and ammonium carbamate buffer can arise differences in the amination reaction of PALs. Our initial aim was to determine which buffer was more suitable for PcPAL. Therefore, we selected these two commonly used buffers as reported in the literature (see: Angew. Chem. Int. Ed. 2014, 53, 4652–4656; Catal. Sci. Technol. 2016(6), 4086-4089; Protein Engineering, Design & Selection, 2010, 23(12): 929-933).”

Besides and after the appearance of the cited manuscripts several studies optimized the reaction medium, ammonia source and ammonia concentration of the PAL mediated reactions (including lit 41,42,53 cited in the manuscript and others), which highlighted that optimal ammonia concentration might vary from substrate to substrate and ranges between 2-4 M carbamate and/or 4-8 M ammonia.

But, nonetheless, to be able to compare the results from the reactions employing different ammonia sources one should try to use at least the same ammonia concentration, which didn't happen in this work. In case if the reaction cannot be repeated in 8 M ammonia or 2 M ammonium carbamate, then **please leave out their comparison** and mention that 5 M ammonia was used as in other studies.

Thanks for your comments. We deleted the comparison part and made changes in the 'Development of a whole-cell biotransformation process' part as following: “To make this process efficient, we optimized different conditions by using β-MeCA as substrate and the commonly used 5 M ammonium hydroxide buffer (NH₄OH buffer, pH 10) the ammonia source.” The corresponding references were also added.

4) short discussion, sentence of the kinetic data should be introduced within the main manuscript, (e.g. above part including Figure)

We described the kinetic data at the end of 'Rational engineering of PcPAL for the amination of β-MeCA' part. The measurement method was also added in the 'Method' part as following: “The kinetic contents were measured in triplicate at 30°C using 96-well UV-microplates. The reactions were carried out in 50 mM PBS buffer (pH 8.0) and the mutant enzyme (PcPAL-L256V-I460V) was maintained at a constant concentration of 0.6 μM, with eight different substrate concentrations ranging from 0.015625 to 1.5 mM. Kinetic measurements were performed using UV spectroscopy to monitor the production of β-MeCA over a 10 min period at a wavelength of 260 nm. Kinetic constants were determined from Michaelis-Menten plots via nonlinear regression analysis.”

Reviewer #3 (Remarks to the Author):

The authors have made changes on the manuscript, but some prior issues remain, particularly regarding the organization, the logic flow, and conclusiveness of the computational modeling. Despite a strong experimental outcome, this research work still appears to be weakened by the lack of hypotheses and justification in data interpretations, leading to apparently unsupported conclusions. To enhance the paper, the author is advised to reconsider what computational studies really matter and make sure the discussions are grounded from molecular theory or physical chemistry basis, rather than discussing most of the data based on their chemical intuition. As such, we provide more specific comments here, with the hope to help the authors further improve the quality, clarity, and impact of the manuscript.

1. The author needs to add a scheme and an introduction about the proposed mechanisms of PcPAL. This will give readers who do not know PAL chemistry a better context of its subsequent discussions.

Thank you for your detailed review and valuable suggestions, which will for sure enhanced the quality of our manuscript. We understand that the absence of a catalytic mechanism leads to logical gaps in the computational section. Therefore, we have added a scheme (see below, now Figure S1) and some explanation about the catalytic mechanism of PcPAL in the beginning of 'In silico analyses of PcPAL towards β -MeCA' part. We believe these additions will strengthen the logic and persuasiveness of the article, enabling readers to gain a more comprehensive understanding of our research work.

2. The docking results cannot support conclusions such as “analogous conformations”. The author should remove them and review them using the conformational ensemble obtained from the cMD simulation.

Thanks for your suggestions. We have thus deleted the description and discussion of docking experiments, as well as the discussion regarding insufficient evidence in the manuscript.

3. The author analyzed “the RMSD of backbone atoms”, “**RMSD** of heavy atoms within 7Å of substrates”, “distances to the R354 residue”, and “the amination reaction distance” based on the cMD simulation. However, the relevance of these metrics to the catalysis is not justified. Replacing the substrate in the MD simulation mainly perturbs the substrate positioning dynamics. But these changes are not directly reflected from those RMSD measurements of the greater protein regions (i.e.: the entire backbone and residues region of 7Å). The measurement of the “distances to the R354 residue” and “the amination reaction distance” comes out a bit sudden without any prior discussion of why these metrics are used. Some discussions appear ungrounded or irrelevant, for example, “ β -MeCA and CA have similar binding modes” and “we could not deduce the reason for the inactivity of PcPAL-WT towards β -MeCA directly from the cMD simulation results”. I suggest removing large part of these irrelevant discussions and keep only the parts that are conclusive from both physical chemistry and biocatalysis perspective.

Thanks for your suggestions. In the revised version, we have removed the description related

to RMSD and made the mechanism of PcPAL more clear. Through clearly defining the key residues for amination reaction, the discussion on “the measurement of the distances to the R354 residue” and “the amination reaction distance” became less abrupt. Additionally, we have deleted some sentences such as “ β -MeCA and CA have similar binding modes” and “we could not deduce the reason for the inactivity of PcPAL-WT towards β -MeCA directly from the cMD simulation results.”

4. The author repetitively uses “the amination reaction distance” in cMD analysis and umbrella sampling. Although the description is intuitive, the physical meaning behind these metrics is vague under Molecular Mechanics context. Calculating binding energy or activation barrier from the umbrella sampling or QM/MM under well-established physics models will be a more legit and natural way to go. Please provide the theoretical justification of how this distance is related to the catalysis. Alternatively, the authors may consider removing most of these discussions if the justification is weak.

The catalytic mechanism shown in Figure S1 highlights the importance of the distances between the active amine and the substrate's α C, as well as between the hydroxyl hydrogen of Tyr110 and the substrate's β C. These distances are crucial indicators of the amination and protonation processes during the reaction. Therefore, we used the computational results of the (CA)/(PcPAL-WT) complex as a reference to understand the inactivity of the (β -MeCA)/(PcPAL-WT) complex. In this context, we revisited the computational section of the manuscript, removed the parts with insufficient evidence (e.g., the IGM part). Additionally, we moved some simulation figures to the Supplementary Information and reduced the discussion in the revised manuscript, focusing only on the differences observed during the simulation, thereby emphasizing the core aspects of the study.

5. Overall, if some computational studies such as umbrella sampling, QM/MM and metrics such as “the amination reaction distance”, “backbone RMSD” cannot be justified, their **removal** may not be a bad choice.

By explaining the catalytic mechanism of PcPAL in the revised manuscript and removing the sections with insufficient evidence and those unrelated to catalysis, the revised manuscript became clearer than the former version.

Reviewer #4 (Remarks to the Author):

Thanks for all your suggestions and comments.

REVIEWER COMMENTS

Reviewer #3 (Remarks to the Author):

We are pleased to find that the authors have addressed most of the comments described in our previous reports. Here are some smaller comments that we believe can help the authors improve the quality of this work.

1. The mechanism schematic has some mistakes. For example, two hydrogen atoms in arginine are missing in all structures, the hydrogen bond should be drawn between O and H instead of the two heavy atoms, the arrow pushing merging two mechanisms in the amination step is misleading, and the use of E1cb and E2 is also misleading. The reaction is the reverse reaction of E1cb and E2. Though existing in the cited paper to describe deamination, we don't suggest using these terms because they appear to mislead the readers to think that an elimination reaction happens. The reaction here is an addition to a double bond, whether the protonation happens in a concerted manner or not does not matter. The manuscript would benefit from modifying the schematic and the corresponding text.

2. Instead of framing the hypothesis as an "overly stable ground state", the authors may consider reframing their hypothesis based on "substrate positioning dynamics". The reason is that the Umbrella Sampling and QM/MM study do not provide any quantitative justification regarding how the substrate is "overly" stabilized. Rather, the metrics such as bonding angle and distance, as discussed in the manuscript, are relevant to the broadly discussed concept of "substrate positioning dynamics". Two fairly recent examples are: J. Phys. Chem. Lett. 2023, 14, 50, 11480–11489; ACS Catal. 2019, 9, 6, 4930–4943. Reframing the hypothesis will make the discussion more convincing.

Reviewer #4 (Remarks to the Author):

Reviewer #3 (Remarks to the Author)

We are pleased to find that the authors have addressed most of the comments described in our previous reports. Here are some smaller comments that we believe can help the authors improve the quality of this work.

Thank you very much for your very positive feedback and for acknowledging our efforts to address the previous comments. We very much appreciate your careful analysis of the 2nd revision and the additional points to improve our manuscript. We of course have made the requested changes according to your comments.

1. The mechanism schematic has some mistakes. For example, two hydrogen atoms in arginine are missing in all structures, the hydrogen bond should be drawn between O and H instead of the two heavy atoms, the arrow pushing merging two mechanisms in the amination step is misleading, and the use of E1cb and E2 is also misleading. The reaction is the reverse reaction of E1cb and E2. Though existing in the cited paper to describe deamination, we don't suggest using these terms because they appear to mislead the readers to think that an elimination reaction happens. The reaction here is an addition to a double bond, whether the protonation happens in a concerted manner or not does not matter. The manuscript would benefit from modifying the schematic and the corresponding text.

Thank you for this very careful analysis of Figure S1. We have revised Figure S1, making the following changes: we added two hydrogen atoms to the Arg residues and illustrated the hydrogen bond between the substrate's oxygen atoms and the hydrogen atoms of Arg. Additionally, to avoid potential confusion for readers, we revised the terms "E1cb" and "E2" in Figure S1 to "reverse reaction of E1cb" and "reverse reaction of E2." Correspondingly, we updated the beginning of the first section in the manuscript as follows: "Currently, the E1cb (via a carbanion intermediate) and the E2 (concerted) mechanisms are widely accepted for PcPAL-mediated deamination reactions. Consequently, the amination reactions catalyzed by PcPAL are likely to proceed through the reverse E1cb and E2 mechanisms (Fig. S1). Despite an ongoing debate between these mechanisms, both of them involve two critical steps in the amination reactions: amination and protonation."

2. Instead of framing the hypothesis as an "overly stable ground state", the authors may consider reframing their hypothesis based on "substrate positioning dynamics". The reason is that the Umbrella Sampling and QMMM study do not provide any quantitative justification regarding how the substrate is "overly" stabilized. Rather, the metrics such as bonding angle and distance, as discussed in the manuscript, are relevant to the broadly discussed concept of "substrate positioning dynamics". Two fairly recent examples are *J. Phys. Chem. Lett.* 2023, 14, 50, 11480–11489; *ACS Catal.* 2019, 9, 6, 4930–4943. Reframing the hypothesis will make the discussion more convincing.

Thank you for your comment. We have revised our hypothesis from "stable ground state" to "substrate positioning dynamics," and the two papers you mentioned have been incorporated into the revised manuscript. The main changes were made in the first and final sections of results part.

Reviewer #4 (Remarks to the Author):

We are also grateful for your co-reviewing of our manuscript as an Early Career Researcher. Your feedback has been very helpful.